# Generic injuries are sufficient to induce ectopic Wnt organizers in *Hydra*

Jack F Cazet, Adrienne Cho, Celina E Juliano*

Department of Molecular and Cellular Biology, University of California, Davis, Davis, United States

**Abstract** During whole-body regeneration, a bisection injury can trigger two different types of regeneration. To understand the transcriptional regulation underlying this adaptive response, we characterized transcript abundance and chromatin accessibility during oral and aboral regeneration in the cnidarian *Hydra vulgaris*. We found that the initial response to amputation at both wound sites is identical and includes widespread apoptosis and the activation of the oral-specifying Wnt signaling pathway. By 8 hr post amputation, Wnt signaling became restricted to oral regeneration. Wnt pathway genes were also upregulated in puncture wounds, and these wounds induced the formation of ectopic oral structures if pre-existing organizers were simultaneously amputated. Our work suggests that oral patterning is activated as part of a generic injury response in *Hydra*, and that alternative injury outcomes are dependent on signals from the surrounding tissue. Furthermore, Wnt signaling is likely part of a conserved wound response predating the split of cnidarians and bilaterians.

*For correspondence:
cejuliano@ucdavis.edu

**Competing interests:** The authors declare that no competing interests exist.

## Introduction

Regeneration is an injury-induced morphogenetic process that enables the restoration of lost or damaged body parts. Although nearly all animals are capable of some form of regeneration, the greatest regenerative capacity is found in the invertebrate species capable of rebuilding their entire body from small tissue fragments through a process called whole-body regeneration. In these highly regenerative systems, amputation injuries trigger morphogenesis on both sides of the amputation plane, leading to the reconstruction of all missing body parts. As a result, the two tissue fragments generated by an amputation will give rise to two morphologically identical individuals, even in cases where the two fragments need to regenerate entirely different structures. This raises the question of how the appropriate morphogenetic programs are activated in response to injury during whole-body regeneration.

Divergent morphogenetic outcomes during regeneration are driven by the differential activation of deeply conserved signaling pathways on either side of the amputation plane (*Cary et al., 2019*; *Holstein et al., 2003*; *Owlarn and Bartscherer, 2016*; *Srivastava et al., 2014*). This results in patterning asymmetries that drive a divergence in gene expression in the two injury sites, leading to the regeneration of different structures. Currently, little is known about how these signaling pathways are reactivated during regeneration or how that reactivation is restricted to only one side of the amputation plane.

The canonical Wnt signaling pathway plays a central role in the positional specification of tissue regenerating along the primary body axis in several distantly related phyla, including cnidarians, acoels, and planarians (*Gurley et al., 2008*; *Hobmayer et al., 2000*; *Lengfeld et al., 2009*; *Petersen and Reddien, 2008*; *Srivastava et al., 2014*; *Stückemann et al., 2017*). In these systems, amputations that transect the primary body axis (i.e., the oral-aboral axis in cnidarians or the anterior-posterior axis in planarians and acoels) induce new Wnt organizer formation on only one side of the amputation plane; in cnidarians, Wnt organizers form only at oral-facing wounds

(*Hobmayer et al., 2000*), and in planarians and acoels, Wnt organizers form only at posterior-facing wounds (*Gurley et al., 2008*; *Petersen and Reddien, 2008*; *Srivastava et al., 2014*). These similarities raise the possibility that injury-induced Wnt organizer formation is an ancestral feature of the metazoan wound response. However, establishing the plausibility of this hypothesis requires a thorough understanding of the regulation and function of injury-induced Wnt signaling in multiple distantly related animal regeneration models.

Wnt signaling has a similar function in specifying posterior tissue during regeneration in both acoels and planarians, but there appear to be marked differences in how the pathway is activated. During acoel regeneration, Wnt signaling is induced via the upregulation of the Wnt ligand *wnt-3* within 3 hr post amputation (hpa; *Ramirez et al., 2020*). This occurs only in posterior-facing injuries and requires the injury-responsive transcription factor (TF) Egr. In contrast, during planarian regeneration several positive regulators of canonical Wnt signaling, including the Wnt ligand *wntP-1*, are initially upregulated as part of an early generic response to all injuries and do not become restricted to posterior regeneration until between 24 and 48 hpa (*Gurley et al., 2010*; *Petersen and Reddien, 2009*; *Stückemann et al., 2017*; *Wenemoser et al., 2012*; *Wurtzel et al., 2015*); however, the secreted Wnt inhibitor *notum*, while also upregulated after all injuries, shows higher levels of upregulation in anterior-facing wounds by 6 hpa (*Petersen and Reddien, 2011*; *Wurtzel et al., 2015*). The TFs involved in upregulating *wntP-1* and *notum* in response to injury in planarians are not known. The differences between acoels and planarians suggest a lack of conservation in the gene regulatory networks that reactivate canonical Wnt signaling during bilaterian regeneration; however, it is unclear if this lack of conservation is the result of divergence from a shared ancestral state or because planarian and acoel regeneration evolved independently.

Cnidaria is the sister clade to bilateria and as such provides an informative context for questions related to the evolution of the metazoan wound response. *Hydra vulgaris* is a well-established cnidarian model that possesses an exceptional regenerative capacity, capable of fully rebuilding its body from tissue pieces containing as few as 300 cells, or ~1% of an adult polyp (*Shimizu et al., 1993*). It has a simple body plan defined by a single oral-aboral axis, with tentacles and a hypostome at the oral end, referred to as the head, and adhesive cells at the aboral end, referred to as the foot (*Figure 1A*). In intact *Hydra*, tissue polarity is dictated by two stable self-maintaining organizers, a head organizer and a foot organizer, that are found at the poles of the oral-aboral axis (*Webster, 1971*). Although the molecular basis for the foot organizer is unknown, the *Hydra* head organizer is based on canonical Wnt signaling; the head organizer expresses multiple Wnt ligands (*Hobmayer et al., 2000*; *Lengfeld et al., 2009*), and Wnt signaling is necessary and sufficient for the specification of oral tissue (*Broun et al., 2005*; *Gee et al., 2010*; *Gufler et al., 2018*). Consistently, regenerating *Hydra* form new Wnt organizers at oral-facing, but not aboral-facing injuries within the first 12 hpa (*MacWilliams, 1983a*; *Petersen et al., 2015*). Currently, there are conflicting data on the function and dynamics of Wnt signaling during cnidarian whole-body regeneration. In addition, the mechanism by which Wnt signaling is activated during regeneration is not well understood. These factors make it difficult to meaningfully compare the injury-induced Wnt signaling activity observed during *Hydra* regeneration to other whole-body regeneration models.

One of the primary points of uncertainty regarding injury-induced Wnt activity in the *Hydra* literature relates to the timing and location of Wnt signaling activation during regeneration. Early grafting experiments in *Hydra* found that donor tissue fragments taken from recent head and foot amputations had an increased capacity to induce ectopic heads in host tissue, suggesting that head-specifying pathways are initially activated regardless of the structure being regenerated (*MacWilliams, 1983a*; *Müller, 1996*). More recently, *Gufler et al., 2018* found that several head-specific transcripts, including *β-catenin*, were initially upregulated during both head and foot regeneration in *Hydra*. They also reported that inhibiting TCF—the downstream transcriptional effector of Wnt signaling—prevented head and foot regeneration, raising the possibility that Wnt signaling may play an active role in both types of regeneration. In a pre-print manuscript, *Wenger et al., 2019* compared transcriptomic data collected during head and foot regeneration and found that transcriptional differences did not arise until between 4 and 8 hpa, indicating that the onset of head-specific transcription occurs several hours after wounding. However, these findings are contrasted by data from *Chera et al., 2009*, who reported that injury-induced apoptosis—which they found occurred only at oral-facing amputations—leads to the activation of Wnt signaling during head but not foot regeneration by 1.5 hpa. These results appear inconsistent with reports of symmetric transcription

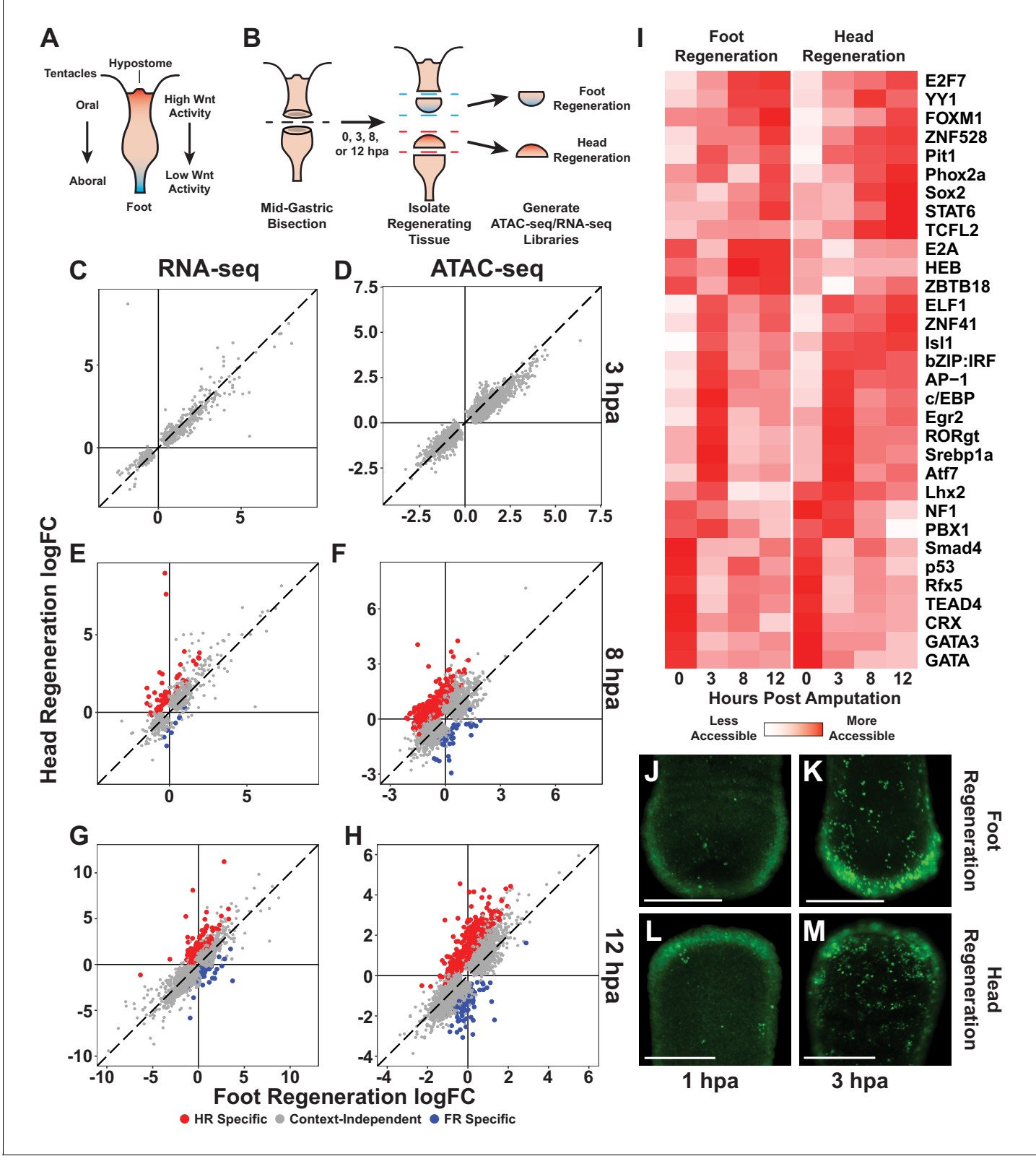

**Figure 1.** The initial transcriptional response to mid-gastric bisection is structure independent but diverges by 8 hr post amputation (hpa). (A) A diagram of the *Hydra* body plan. Red coloration indicates a head-specific molecular signature. Blue coloration indicates a foot-specific molecular signature. (B) Experimental design for ATAC-seq and RNA-seq library generation. (C–H) Comparison of average $\log_2$ fold change ($\log_2$FC) in transcript abundance (C, E, G) or chromatin accessibility (D, F, H) during head and foot regeneration at 3, 8, and 12 hpa. The transcriptional

*Figure 1 continued on next page*

*Figure 1 continued*

response during head and foot regeneration is identical at 3 hpa but becomes distinct by 8 hpa. Features that showed significant differences in the injury response during head and foot regeneration when compared to 0 hpa controls are highlighted in red for head regeneration-specific features or blue for foot regeneration-specific features (false discovery rate [FDR] ≤ 1e-3 for RNA-seq and FDR ≤ 1e-4 for ATAC-seq). The dotted line indicates perfect correlation between regeneration types. All significance values were calculated with edgeR using quasi-likelihood tests of count data fitted to a negative binomial generalized log-linear model. Information on RNA-seq and ATAC-seq biological replicates is provided in *Tables 1* and *2*. Individual FDR values for all genewise/peakwise tests can be found in *Supplementary files 1* and *2*. (I) Heatmap of the average relative accessibility of transcription factor binding motifs during regeneration in head and foot regenerates calculated using chromVAR. Transcription factor activity during the first 12 hr of regeneration was highly dynamic and largely similar between head and foot regeneration. (J–M) Late-stage apoptotic cells labeled in *Hydra* undergoing head or foot regeneration using acridine orange. Widespread apoptosis was observed during head and foot regeneration at 3 hpa but was largely absent at 1 hpa. Additional representative images are presented in *Figure 1—figure supplement 4*. Scale bars indicate 250 µm.

The online version of this article includes the following source data and figure supplement(s) for figure 1:

**Source data 1.** Excel workbook containing all tables that were used to generate plots for *Figure 1C–I*.
**Source data 2.** Transcription factor binding motifs included in this study.
**Figure supplement 1.** Sample ATAC-seq data centered on the *wnt3* locus.
**Figure supplement 2.** Sample ATAC-seq data centered on the *wnt9/10c* locus.
**Figure supplement 3.** Sample ATAC-seq data centered on the *pitx* locus.
**Figure supplement 4.** Oral and aboral amputations induce apoptosis within 3 hr post amputation (hpa).

during early head and foot regeneration as this would require that the dramatic asymmetries between the two types of regeneration at 1.5 hpa are not associated with any concomitant transcriptional changes.

The available data on *Hydra* regeneration leave significant gaps in knowledge that make it difficult to effectively understand the mechanisms underlying patterning during regeneration. In particular, the role of canonical Wnt signaling during *Hydra* regeneration remains incompletely characterized. Furthermore, it is unclear when and how canonical Wnt signaling genes, particularly Wnt ligands and negative regulators of Wnt signaling, are upregulated during head and foot regeneration following injury. We therefore sought to gain insight into these questions by comprehensively characterizing changes in chromatin accessibility and transcript abundance during the first 12 hr of head and foot regeneration—the period when new organizers are being formed. We found that the initial transcriptional response was indistinguishable during head and foot regeneration and included the symmetric upregulation of apoptotic markers and canonical Wnt signaling components. This was correlated with a symmetric increase in chromatin accessibility near predicted TCF binding sites, suggesting that Wnt signaling was initially activated during both head and foot regeneration. By 8 hpa, head and foot regeneration became transcriptionally distinct, which coincided with Wnt signaling transcripts becoming restricted to head regeneration. Inhibiting TCF delayed this transcriptional divergence genome-wide, suggesting a central role for canonical Wnt signaling during oral and aboral regeneration. A systematic analysis of chromatin accessibility data found that injury-induced Wnt component upregulation was likely driven by conserved injury-responsive basic leucine zipper (bZIP) TFs. In support of a link between the generic injury response and canonical Wnt signaling, we found that non-amputation injuries also triggered the upregulation of Wnt pathway components and could induce ectopic head formation when pre-existing organizers were removed. In addition, prolonged aboral-facing amputation injuries also induced ectopic head formation, but only after inhibitory signals from head-regenerating tissue were removed. Overall, these findings suggest that injuries initiate an oral patterning cascade regardless of the surrounding tissue context through the activation of canonical Wnt signaling, and that the outcome of this activation is regulated by long-range signals generated by organizers. These results also suggest that the upregulation of canonical Wnt signaling components may be part of an ancestral metazoan injury response.

## Results

### The initial transcriptional response to injury does not depend on the structure being regenerated

To better understand the transcriptional regulatory mechanisms underlying positional divergence during head and foot regeneration in *Hydra*, we generated ATAC-seq (*Buenrostro et al., 2013*;

*Corces et al., 2017*) and RNA-seq libraries from polyps undergoing head and foot regeneration at 0, 3, 8, and 12 hr after mid-gastric bisection (*Figure 1B*, *Figure 1—figure supplements 1–3*, *Tables 1* and *2*). In designing our experiments, we focused on the first 12 hr of regeneration because new organizers are established by approximately 12 hpa (*Hicklin and Wolpert, 1973*; *MacWilliams, 1983a*). ATAC-seq reveals accessible regions of chromatin, allowing us to identify active cis-regulatory elements genome-wide. In addition, comparing ATAC-seq data from different timepoints allows us to characterize changes in cis-regulatory element activity over the course of regeneration. The inclusion of RNA-seq data allows us to correlate changes in chromatin accessibility to changes in transcript abundance in nearby genes. Thus, these data enable the characterization of gene regulatory network activity during head and foot regeneration that started from the same initial injury.

We first sought to identify the earliest transcriptional differences that arose between head and foot regeneration. To do this, we identified transcripts and ATAC-seq peaks that changed in significantly different ways during the two types of regeneration (full differential gene expression results are available in *Supplementary files 1* and *2*). At 3 hpa, although we detected extensive changes in chromatin accessibility and transcript abundance relative to uninjured controls, there were no significant differences in those changes when comparing head and foot regeneration (*Figure 1C, D*). However, by 8 hpa there was clear evidence of head and foot regeneration-specific transcriptional responses, with 63 transcripts and 464 peaks showing significant differences when comparing head and foot regeneration (*Figure 1E, F*). The number of transcriptional differences increased at 12 hpa, with 139 transcripts and 631 peaks exhibiting differential activation during the two types of regeneration (*Figure 1G, H*). We postulate that we observe a greater number of differentially activated peaks relative to transcripts because gene loci are often associated with multiple peaks. In addition, changes in certain individual cis-regulatory elements may not be sufficient to drive detectable differences in transcript abundance. Overall, our analysis demonstrates that the initial transcriptional response to amputation in *Hydra* does not depend on the structure being regenerated, and that transcriptional differences between head and foot regeneration begin to arise between 3 and 8 hpa.

To evaluate the congruence of our ATAC-seq and RNA-seq data, we used a hypergeometric enrichment test to determine if ATAC-seq peaks that were identified as head or foot regeneration-specific were significantly enriched near head or foot regeneration-specific gene loci. We found that structure-specific transcripts were significantly enriched in nearby structure-specific peaks (14.1-fold enrichment at 8 hpa, p=3.6e-25; 9.8-fold enrichment at 12 hpa, p=3.7e-40), demonstrating a meaningful correspondence between our ATAC-seq and RNA-seq datasets. We also compared our RNA-seq data with a previously released transcriptomic dataset of *Hydra* head and foot regeneration and found that there was a significant overlap in the head and foot regeneration-specific transcripts identified in the two studies (507-fold enrichment at 8 hpa, p~0; 35-fold enrichment at 12 hpa, p=1.11e-93) (*Wenger et al., 2019*). We therefore found that the identification of transcriptional asymmetries during regeneration was highly reproducible across independent studies and across orthogonal next-generation sequencing methodologies.

We next used our chromatin accessibility data to identify candidate TFs that could plausibly underlie the observed transcriptional changes during regeneration. To do this, we used chromVAR (*Schep et al., 2017*) to systematically identify transcription factor binding motifs (TFBMs) throughout the genome and characterize how the aggregate chromatin accessibility near those motifs changed over the course of regeneration. Using this approach, we identified 32 TFBMs that were associated with significant variability in chromatin accessibility during the first 12 hr of regeneration (*Figure 1I*; TFBM sequences are provided in *Figure 1—source data 2*). Consistent with our observation that the majority of transcriptional changes during head and foot regeneration were highly similar, we found that the large majority of injury responsive TFBMs had similar accessibility dynamics during oral and aboral regeneration. These data suggest that injury-responsive gene regulatory networks behave in a largely symmetric manner during early head and foot regeneration. Notably, we found that the TFBMs associated with increases in accessibility from 0 to 3 hpa included motifs associated with TFs known to play a role in the early injury response in bilaterians such as AP-1, Atf, and Egr (*Schäfer and Werner, 2007*). The likely involvement of these TFs in *Hydra* regeneration suggests that they have an ancestral role in the metazoan wound response that predates the split of cnidaria and bilateria.

**Table 1.** ATAC-seq library statistics.

A total of 71 ATAC-seq libraries were generated, with between 3 and 5 biological replicates per treatment. HPA refers to hours post amputation. + in the iCRT column indicates regenerating animals were pre-incubated in 5 μM iCRT14 for 2 hr prior to amputation and then left in the iCRT14 solution until tissue was collected for library preparation; – in the iCRT column indicates animals were left untreated. Total read pairs refers to the number of raw read pairs generated for each library. Final mapped read pairs refers to the number of read pairs remaining after mitochondrial, duplicated, and unmapped or ambiguously mapped reads were removed. Transcription start sites (TSS) enrichment refers to the fold enrichment in ATAC-seq signal at the TSS of 2000 highly expressed genes relative to regions ±1 kb from the TSS. Reproducible peaks refers to the number of peaks within each treatment group that were biologically reproducible in at least three pairwise comparisons using an irreproducible discovery rate cutoff of 0.1. Benchmarks for both the TSS score and the number of reproducible peaks are highly dependent on the model system and the quality of the reference annotation, but in well-studied systems such as mice, ENCODE considers TSS scores > 5 and >50,000,000 reproducible peaks to be acceptable. The self-consistency ratio refers to the largest fold difference in the number of reproducible peaks recovered from pseudo-replicates (see Materials and methods) when comparing biological replicates within a single treatment group. The rescue ratio refers to the fold difference in the number of reproducible peaks recovered from a dataset with perfect reproducibility (generated by pooling and then randomly splitting reads from a given treatment group) compared to the number of true biologically reproducible peaks. High-quality datasets should have self-consistency ratios and rescue ratios <2.

| Sample ID | HPA | iCRT | Total read pairs | Final mapped read pairs | TSS enrichment | Reproducible peaks | Self-consistency ratio | Rescue ratio |
|---|---|---|---|---|---|---|---|---|
| 0F1 | 0 | – | 68865654 | 31845047 | 6.15 | 78158 | 1.17 | 1.00 |
| 0F2 | 0 | – | 65301503 | 21970963 | 7.16 | | | |
| 0F3 | 0 | – | 80642153 | 25843928 | 6.61 | | | |
| 0F4 | 0 | – | 76300368 | 25925595 | 6.79 | | | |
| 0F5 | 0 | – | 81363105 | 20714765 | 6.79 | | | |
| 0H1 | 0 | – | 72424905 | 31676679 | 6.31 | 82396 | 1.17 | 1.04 |
| 0H2 | 0 | – | 68870186 | 27376967 | 6.81 | | | |
| 0H3 | 0 | – | 81121398 | 31084966 | 6.50 | | | |
| 0H4 | 0 | – | 67378757 | 26076124 | 6.58 | | | |
| 0H5 | 0 | – | 81380944 | 26247390 | 6.44 | | | |
| 3F1 | 3 | – | 120749030 | 35747201 | 6.44 | 57980 | 1.11 | 1.20 |
| 3F2 | 3 | – | 133453367 | 37551124 | 6.85 | | | |
| 3F3 | 3 | – | 128451210 | 44807082 | 6.76 | | | |
| 3H1 | 3 | – | 127697432 | 33201124 | 7.45 | 58259 | 1.09 | 1.15 |
| 3H2 | 3 | – | 124784335 | 38517160 | 7.09 | | | |
| 3H3 | 3 | – | 130420684 | 36703497 | 7.05 | | | |
| 8F1 | 8 | – | 71642160 | 20270917 | 7.30 | 84248 | 1.29 | 1.05 |
| 8F2 | 8 | – | 77325666 | 17968245 | 7.83 | | | |
| 8F3 | 8 | – | 75894343 | 17391339 | 7.15 | | | |
| 8F4 | 8 | – | 71117490 | 25653837 | 6.98 | | | |
| 8F5 | 8 | – | 74203082 | 30614424 | 7.17 | | | |
| 8H1 | 8 | – | 72411934 | 18714649 | 7.18 | 83974 | 1.18 | 1.04 |
| 8H2 | 8 | – | 76210411 | 16618719 | 7.78 | | | |
| 8H3 | 8 | – | 74495918 | 22207972 | 6.53 | | | |
| 8H4 | 8 | – | 68874107 | 28397085 | 6.78 | | | |
| 8H5 | 8 | – | 85121623 | 26333487 | 6.31 | | | |
| 12F1 | 12 | – | 35989738 | 16371364 | 7.81 | 71949 | 1.09 | 1.00 |
| 12F2 | 12 | – | 37815306 | 16333732 | 7.26 | | | |
| 12F3 | 12 | – | 31696505 | 10175518 | 7.47 | | | |
| 12F4 | 12 | – | 35835498 | 10383772 | 8.43 | | | |
| 12F5 | 12 | – | 27753392 | 9458838 | 7.78 | | | |

*Table 1 continued on next page*

*Table 1 continued*

| Sample ID | HPA | iCRT | Total read pairs | Final mapped read pairs | TSS enrichment | Reproducible peaks | Self-consistency ratio | Rescue ratio |
|---|---|---|---|---|---|---|---|---|
| 12H1 | 12 | – | 46530322 | 17225290 | 6.98 | 71463 | 1.16 | 1.05 |
| 12H2 | 12 | – | 33411133 | 16231389 | 6.68 | | | |
| 12H3 | 12 | – | 38128076 | 9053148 | 8.13 | | | |
| 12H4 | 12 | – | 37038785 | 8069733 | 7.94 | | | |
| 12H5 | 12 | – | 35508148 | 9077295 | 7.10 | | | |
| 0iF1 | 0 | + | 29450456 | 13933591 | 6.99 | 68393 | 1.13 | 1.03 |
| 0iF2 | 0 | + | 47282308 | 23905822 | 6.18 | | | |
| 0iF3 | 0 | + | 54230058 | 25344368 | 6.02 | | | |
| 0iF4 | 0 | + | 40244798 | 18954394 | 6.43 | | | |
| 0iH1 | 0 | + | 64726838 | 26952996 | 5.79 | 70771 | 1.08 | 1.05 |
| 0iH2 | 0 | + | 46838471 | 22246978 | 6.8 | | | |
| 0iH3 | 0 | + | 40909454 | 18903370 | 6.87 | | | |
| 0iH4 | 0 | + | 54067924 | 25119346 | 6.35 | | | |
| 3iF1 | 3 | + | 54065239 | 10886919 | 5.96 | 67391 | 1.16 | 1.06 |
| 3iF2 | 3 | + | 48714606 | 14470686 | 6.55 | | | |
| 3iF3 | 3 | + | 44688328 | 16294810 | 6.71 | | | |
| 3iF4 | 3 | + | 36970695 | 15737834 | 5.81 | | | |
| 3iF5 | 3 | + | 32077041 | 12065542 | 5.3 | | | |
| 3iH1 | 3 | + | 46903548 | 9057427 | 7.19 | 62579 | 1.12 | 1.00 |
| 3iH2 | 3 | + | 41743618 | 15654587 | 6.79 | | | |
| 3iH3 | 3 | + | 38888634 | 12484017 | 6.85 | | | |
| 3iH4 | 3 | + | 45598782 | 16242466 | 6.54 | | | |
| 8iF1 | 8 | + | 50536148 | 13430112 | 6.78 | 61139 | 1.26 | 1.05 |
| 8iF2 | 8 | + | 54337061 | 12648440 | 7.11 | | | |
| 8iF3 | 8 | + | 52282170 | 11852948 | 5.21 | | | |
| 8iF4 | 8 | + | 42891267 | 12120423 | 5.43 | | | |
| 8iH1 | 8 | + | 42290670 | 11796387 | 6.72 | 65043 | 1.57 | 1.10 |
| 8iH2 | 8 | + | 39596909 | 14998918 | 5.20 | | | |
| 8iH3 | 8 | + | 39883594 | 12563437 | 5.27 | | | |
| 8iH4 | 8 | + | 28249292 | 8393601 | 4.57 | | | |
| 8iH5 | 8 | + | 43953185 | 10799451 | 7.85 | | | |
| 12iF1 | 12 | + | 39257829 | 13879417 | 7.14 | 70505 | 1.05 | 1.01 |
| 12iF2 | 12 | + | 42214218 | 16259721 | 7.39 | | | |
| 12iF3 | 12 | + | 40921039 | 10536561 | 7.64 | | | |
| 12iF4 | 12 | + | 45775901 | 12512902 | 7.54 | | | |
| 12iF5 | 12 | + | 36779064 | 10269191 | 7.39 | | | |
| 12iH1 | 12 | + | 43772456 | 18806981 | 6.49 | 65492 | 1.14 | 1.01 |
| 12iH2 | 12 | + | 37690946 | 15270494 | 6.95 | | | |
| 12iH3 | 12 | + | 42788391 | 11435578 | 7.54 | | | |
| 12iH4 | 12 | + | 42138119 | 15016298 | 6.48 | | | |

**Table 2.** RNA-seq library statistics.

A total of 42 RNA-seq libraries were generated, with three biological replicates per treatment. HPA refers to hours post amputation. + in the iCRT column indicates regenerating animals were pre-incubated in 5 µM iCRT14 for 2 hr prior to amputation and then left in the iCRT14 solution until tissue was collected for library preparation; – in the iCRT column indicates animals were left untreated. Total reads refers to the number of raw reads generated for each library. Final mapped reads refers to the number of reads that were successfully mapped to the *Hydra* 2.0 genome gene models.

| Sample ID | HPA | iCRT | Total reads | Final mapped reads |
|---|---|---|---|---|
| 0F1 | 0 | – | 29284004 | 21541719 |
| 0F2 | 0 | – | 24597360 | 17966439 |
| 0F3 | 0 | – | 23926341 | 17350620 |
| 0H1 | 0 | – | 25833101 | 18858726 |
| 0H2 | 0 | – | 27241258 | 19701074 |
| 0H3 | 0 | – | 26803090 | 19487287 |
| 3F1 | 3 | – | 24824468 | 18479637 |
| 3F2 | 3 | – | 23840505 | 17694124 |
| 3F3 | 3 | – | 23900532 | 17743737 |
| 3H1 | 3 | – | 25874570 | 19207371 |
| 3H2 | 3 | – | 24172310 | 17898155 |
| 3H3 | 3 | – | 23397937 | 17670977 |
| 8F1 | 8 | – | 23040344 | 17160851 |
| 8F2 | 8 | – | 26019557 | 19216264 |
| 8F3 | 8 | – | 29378123 | 22033550 |
| 8H1 | 8 | – | 25548801 | 19003624 |
| 8H2 | 8 | – | 21601150 | 16154188 |
| 8H3 | 8 | – | 23904011 | 17656019 |
| 12F1 | 12 | – | 26995204 | 18138493 |
| 12F2 | 12 | – | 23703527 | 15445837 |
| 12F3 | 12 | – | 25488245 | 16491319 |
| 12H1 | 12 | – | 34878154 | 22910100 |
| 12H2 | 12 | – | 31662571 | 21545505 |
| 12H3 | 12 | – | 28192288 | 18744249 |
| 0iF1 | 0 | – | 26635485 | 17851347 |
| 0iF2 | 0 | + | 28052169 | 18904815 |
| 0iF3 | 0 | + | 30718370 | 20708864 |
| 0iH1 | 0 | + | 36328076 | 23784082 |
| 0iH2 | 0 | + | 26044439 | 17240456 |
| 0iH3 | 0 | + | 31384073 | 20624671 |
| 8iF1 | 8 | + | 37979732 | 24369941 |
| 8iF2 | 8 | + | 31722061 | 20432162 |
| 8iF3 | 8 | + | 38252155 | 25503172 |
| 8iH1 | 8 | + | 34522120 | 22339316 |
| 8iH2 | 8 | + | 34442705 | 21537194 |
| 8iH3 | 8 | + | 29727022 | 19082280 |
| 12iF1 | 12 | + | 22354289 | 13460197 |
| 12iF2 | 12 | + | 35141637 | 21948916 |
| 12iF3 | 12 | + | 35240879 | 22241372 |

*Table 2 continued on next page*

*Table 2 continued*

| Sample ID | HPA | iCRT | Total reads | Final mapped reads |
|-----------|-----|------|-------------|--------------------|
| 12iH1 | 12 | + | 25883305 | 16660844 |
| 12iH2 | 12 | + | 32237790 | 20843729 |
| 12iH3 | 12 | + | 73113728 | 44132834 |

## Mid-gastric bisection induces extensive apoptosis in oral and aboral-facing wounds by 3 hpa

Although our finding of an initially symmetric transcriptional response during head and foot regeneration is consistent with other transcriptional studies in *Hydra* (*Gufler et al., 2018*; *Wenger et al., 2019*), it appears to be inconsistent with a previous report of extensive apoptosis occurring only at oral-facing amputations by 1 hpa (*Chera et al., 2009*). Apoptosis induces dramatic changes in gene expression (*Sun et al., 2017*), and asymmetric apoptosis would therefore be expected to result in dramatic transcriptional asymmetries during early head and foot regeneration. To reconcile these seemingly discrepant observations, we sought to characterize apoptosis during early head and foot regeneration using acridine orange, a previously validated stain for late-stage apoptotic cells in *Hydra* (*Cikala et al., 1999*; *Kuznetsov et al., 2002*; *Technau et al., 2003*). In contrast to previous findings, we found no evidence of head regeneration-specific apoptosis at 1 hpa (*Figure 1J, L*, *Figure 1—figure supplement 4A*). Rather, at this timepoint both head and foot regeneration exhibited limited apoptosis and appeared largely similar to uninjured controls. However, by 3 hpa, we observed extensive apoptosis throughout the body in both head and foot regenerating tissue fragments (*Figure 1K, M*, *Figure 1—figure supplement 4A*). Consistent with this observation, our transcriptomic data showed that numerous apoptotic markers such as *caspase-3c*, *bcl-like-6* (*Lasi et al., 2010*), *p53*, and *bax* were upregulated in a similar fashion during head and foot regeneration at 3 hpa (*Figure 1—figure supplement 4B–I*; the *Hydra* 2.0 genome gene model IDs for the gene names used throughout this study can be found in *Table 3*). These data indicate that amputations induce apoptosis in a symmetric manner at both oral and aboral-facing injuries by 3 hpa, consistent with a generic initial response to injury. This conclusion is also consistent with the findings reported by *Tursch et al., 2020*, who found no evidence of head regeneration-specific apoptosis at 2 hpa following mid-gastric bisection.

## Wnt signaling is likely activated during early head and foot regeneration, but becomes head specific by 8 hpa

For regeneration to rebuild oral and aboral structures, part of the injury response must include the reactivation of the pathways that specify and maintain these structures under steady-state conditions. Therefore, to better understand how transcriptional symmetry is broken during regeneration, we looked for evidence of the reactivation of head and foot-specific pathways to identify the genes involved in the early stages of new head or foot organizer formation. This required that we first identify genes associated with uninjured head and foot tissue. To do this, we used a previously published single-cell RNA-seq atlas (*Siebert et al., 2019*) to identify genes that showed significantly different expression when comparing epithelial head cell clusters to epithelial foot cell clusters. We then characterized changes in the transcript abundance of these genes along with changes in chromatin accessibility in nearby peaks during regeneration.

At 8 hpa, the earliest time point in our dataset with transcriptional differences between head and foot regeneration, there were 55 transcripts and 426 peaks that showed significantly higher activation during head regeneration relative to foot regeneration. We found that 23/55 of the head regeneration-specific transcripts also exhibited head-specific expression in uninjured polyps, and 65/426 of the head regeneration-specific peaks were located near genes that exhibited head-specific expression in uninjured polyps (*Figure 2A, B*). These early head-specific factors included previously characterized genes associated with the *Hydra* head such as *wnt9/10c* (*Lengfeld et al., 2009*), *wnt3* (*Hobmayer et al., 2000*), *brachyury1* (*Technau and Bode, 1999*), *naked cuticle* (*Petersen et al., 2015*), and *sp5* (*Vogg et al., 2019*). In contrast, we did not observe a similar activation of foot-specific transcription at 8 hpa, although there was some evidence of foot-specific chromatin remodeling

**Table 3.** Genome gene model IDs for in-text gene names.

The *Hydra* 2.0 genome gene model IDs that correspond to the gene names used in this study. The *Hydra* 2.0 genome gene models can be found at arusha.nhgri.nih.gov/hydra/download/?dl=n.

| Gene name | Genome gene model ID |
|---|---|
| *axin* | g14938 |
| *bax* | g7775 |
| *bcl-like-6* | g22655 |
| *brachyury1* | g24952 |
| *budhead* | g24126 |
| *caspase3-like* | g664 |
| *caspase-3c* | g31854 |
| *cr3l* | g32585 |
| *creb* | g16491 |
| *dapk2-like* | g14048 |
| *dff40* | g11021 |
| *dishevelled* | g29781 |
| *distal-less* | g18245 |
| *fos* | g23720 |
| *foxd2-like* | g28449 |
| *fzd4/9/10* | g27891 |
| *gata-3-like* | g20911 |
| *gremlin-like* | g14229 |
| *hsp70-like* | g16168 |
| *jun* | g1450 |
| *naked cuticle* | g30442 |
| *nk-2* | g31954 |
| *notum* | g26902 |
| *prdl-a* | g15226 |
| *p53* | g21693 |
| *sFRP* | g12274 |
| *sp5* | g33422 |
| *tcf* | g27364 |
| *wnt3* | g28064 |
| *wnt7* | g29750 |
| *wnt9/10c* | g33373 |
| *wntless* | g18842 |

(*Figure 2A, B*). At 12 hpa, however, several genes enriched in homeostatic foot tissue, including the known foot-associated TFs *distal-less* (*Hemmrich et al., 2012*) and *nk-2* (*Grens et al., 1996*; *Figure 2C, D*), were upregulated in a foot regeneration-specific manner. We also found two previously undescribed foot-specific TFs, *gata-3-like* and *foxd2-like*, that were upregulated during foot regeneration by 12 hpa. We hypothesize that these early context-specific genes reside at the top of the regulatory hierarchy directing the specification of head or foot tissue during regeneration.

We then used chromVAR to identify candidate TFBMs that could plausibly be involved in the onset of head and foot regeneration-specific transcription by looking for TFBMs associated with asymmetric changes in chromatin accessibility. We found that the TFBM for TCF, the downstream transcriptional effector of canonical Wnt signaling, was among the small number of motifs associated with asymmetric accessibility, exhibiting head regeneration-specific increases by 8 hpa onward

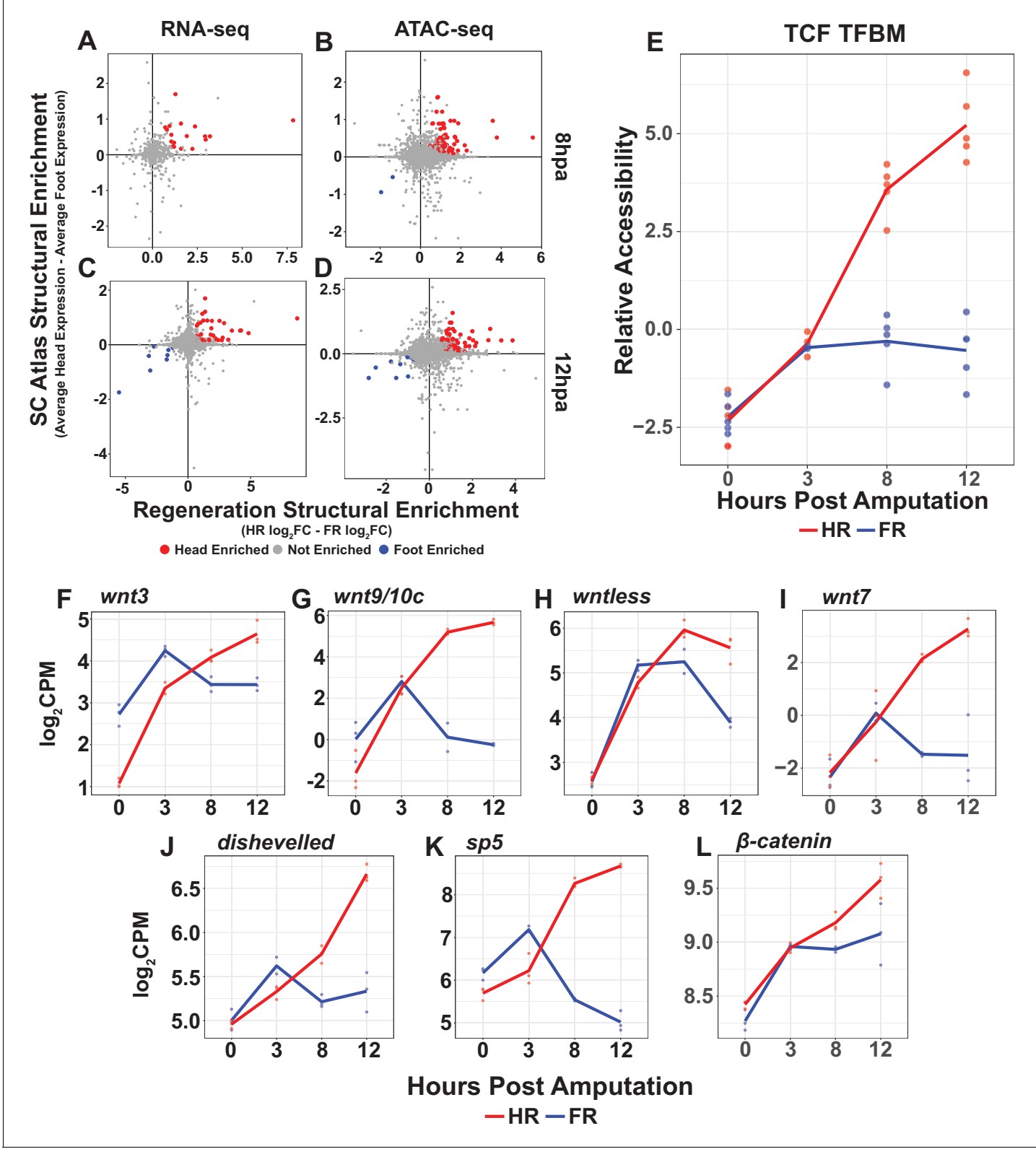

**Figure 2.** Wnt signaling is likely initially activated in a structure-independent manner but becomes restricted to regenerating head tissue by 8 hr post amputation (hpa). (A–D) Plots comparing enrichment in head and foot tissue in uninjured *Hydra* and during regeneration. Genes associated with the homeostatic maintenance of head and foot organizers are upregulated within 12 hpa. Enrichment in uninjured head and foot tissue (i.e., single-cell [SC] Atlas Structural Enrichment) was calculated by comparing the average relative expression levels in epithelial head cells isolated from the *Hydra* SC atlas

*Figure 2 continued on next page*

*Figure 2 continued*

to expression in epithelial foot cells. A positive fold change indicates enrichment in head tissue, and a negative fold change indicates enrichment in foot tissue. Head or foot-specific activation during regeneration (i.e., Regeneration Context Effect) was calculated by subtracting the $\log_2$FC in regenerating head tissue from the log2FC in regenerating foot tissue using the 0 hpa timepoint as the control. A positive fold change indicates enrichment in head-regenerating tissue, and a negative fold change indicates enrichment in foot-regenerating tissue. Transcripts (**A, C**) or ATAC-seq peaks (**B, D**) that were enriched in head-regenerating tissue and were also enriched in uninjured head tissue are highlighted in red. ATAC-seq peaks or transcripts that were enriched in foot-regenerating tissue and were also enriched in uninjured foot tissue are highlighted in blue. (**E**) Average relative chromatin accessibility plot of ATAC-seq peaks containing the TCF transcription factor binding motif (TFBM) during head and foot regeneration. The TCF TFBM is associated with increases in accessibility during both head and foot regeneration at 3 hpa, but subsequent increases in accessibility are restricted to head regeneration. Relative changes in chromatin accessibility were calculated using chromVAR using the HOMER TCFL2 TFBM sequence. (**F–L**) RNA expression plots showing average normalized RNA-seq read counts for Wnt signaling components in $\log_2$ counts per million ($\log_2$CPM) during head and foot regeneration. Wnt signaling components are initially upregulated during head and foot regeneration but become head regeneration-specific by 12 hpa. HR: head regeneration; FR: foot regeneration.

The online version of this article includes the following source data and figure supplement(s) for figure 2:

**Source data 1.** Excel workbook containing all tables that were used to generate plots for *Figure 2A–D*.
**Figure supplement 1.** Canonical Wnt signaling components are upregulated during foot regeneration.
**Figure supplement 1—source data 1.** Excel workbook containing the table used to generate the heatmap in *Figure 2—figure supplement 1*.

(*Figure 1I*, *Figure 2E*). Because TCF constitutively binds to target loci and activates transcription in part by recruiting chromatin remodeling enzymes that increase chromatin accessibility (*Cadigan, 2012*), we interpret these data as indicative of a head regeneration-specific increase in TCF transcriptional activation. Thus, our analysis suggests that canonical Wnt signaling is activated in a head regeneration-specific manner by 8 hpa and is among the earliest asymmetries in gene regulatory network activity during regeneration, consistent with its important role in specifying oral tissue (*Broun et al., 2005*; *Gee et al., 2010*; *Hobmayer et al., 2000*; *Lengfeld et al., 2009*).

While examining the chromatin accessibility dynamics associated with TCF, we noted that although chromatin accessibility near TCF TFBMs increased in a head regeneration-specific manner from 8 hpa onward, at 3 hpa the TCF TFBM was associated with symmetric accessibility increases during both head and foot regeneration (*Figure 2E*). This raised the possibility that canonical Wnt signaling may be activated as part of the initial symmetric response to injury. To further explore the plausibility of this hypothesis, we used KEGG pathway annotations to systematically identify and characterize the expression dynamics of canonical Wnt pathway components during regeneration (*Kanehisa and Goto, 2000*). We found that several canonical Wnt signaling components—including *wnt9/10c*, *wnt3*, *wnt7*, *wntless*, *dishevelled*, *β-catenin*, and *sp5*—were first upregulated during the initial symmetric phase of regeneration and did not become head regeneration specific until 8 or 12 hpa (*Figure 2F–L*, *Figure 2—figure supplement 1*). These observations demonstrate that, similar to the planarian wound response (*Gurley et al., 2010*; *Petersen and Reddien, 2009*; *Stückemann et al., 2017*; *Wurtzel et al., 2015*), canonical Wnt signaling components are upregulated as part of the early generic injury response in *Hydra*. In addition, it suggests that the upregulation of these genes initially results in the activation of Wnt signaling during both head and foot regeneration, but that Wnt signaling becomes restricted to head regeneration by 8 hpa.

## TCF inhibition delays the onset of head and foot-specific transcription during regeneration transcriptome-wide

The possible activation of Wnt signaling during both head and foot regeneration raises questions about the pathway's function during regeneration. Wnt signaling has a well-established function specifying oral tissue during regeneration (*Broun et al., 2005*; *Gufler et al., 2018*; *Lengfeld et al., 2009*), but the role of Wnt signaling during aboral regeneration is poorly understood. Recent work by *Gufler et al., 2018* found that the small molecule inhibitor iCRT14, which both inhibits the interaction between β-catenin and TCF and disrupts TCF binding to DNA (*Gonsalves et al., 2011*), blocked head and foot regeneration and prevented the downregulation of several head-specific transcripts during foot regeneration; however, because this study only characterized a small number of genes, the broader function of TCF in regulating transcription during regeneration remains incompletely understood. We therefore sought to characterize the impact of TCF inhibition during

regeneration across the entire transcriptome by generating RNA-seq and ATAC-seq libraries from head and foot regenerating tissue fragments treated with iCRT14.

Before performing the regeneration timecourse experiments, we first validated that our iCRT14 treatment conditions yielded results consistent with previously published findings (*Gufler et al., 2018*). To do this, we tested the ability of iCRT14 to inhibit head and foot regeneration. We found

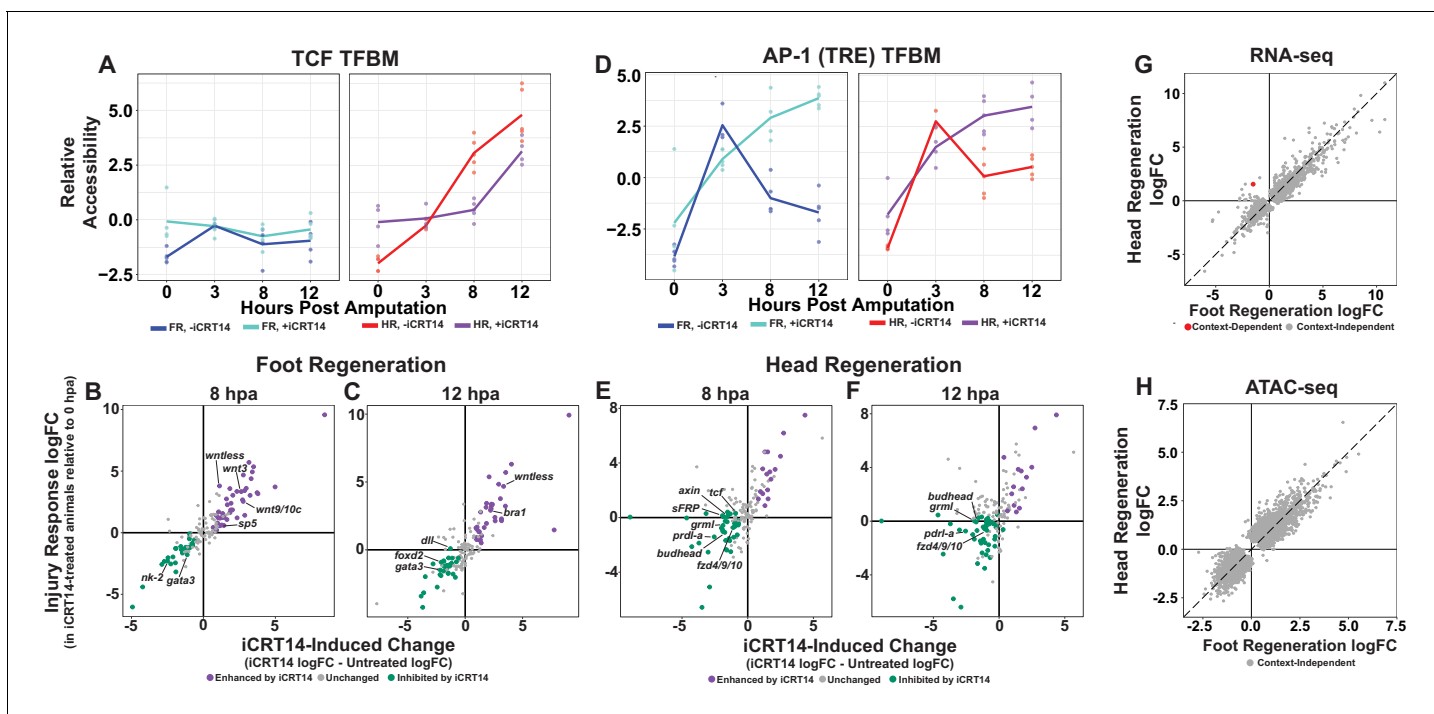

**Figure 3.** TCF is required for the initiation of head and foot-specific transcription during regeneration. (**A**) Average relative chromatin accessibility plot of peaks containing the TCF transcription factor binding motif (TFBM) during head and foot regeneration in both untreated animals and animals incubated in 5 µM iCRT14. iCRT14 treatment significantly diminished and delayed increases in chromatin accessibility associated with the TCF TFBM during head regeneration. The HOMER TCFL2 motif sequence was used to identify putative TCF binding sites. (**B, C**) Scatter plots depicting the effect of 5 µM iCRT14 during foot regeneration on the injury induced expression of genes that were specific to head or foot regeneration in untreated animals. TCF inhibition prevents the upregulation of foot regeneration-specific genes and prolongs the expression of head-specific genes during foot regeneration. Negative values on the x-axis indicate a decrease in the injury-induced expression of a gene in iCRT14-treated animals relative to untreated controls, while positive values indicate an increase in injury-induced expression. Negative values on the y-axis indicate that the gene was downregulated relative to 0 hr post amputation (hpa) controls in iCRT14-treated animals, while positive values indicate that the gene was upregulated. Green coloration denotes transcripts that showed diminished injury-induced expression in iCRT14-treated animals (i.e., expression was TCF-dependent). Purple coloration denotes transcripts that showed enhanced injury-induced expression in iCRT14-treated animals (i.e., expression was inhibited by TCF). TCF-dependent transcripts (green) were defined as transcripts that were not significantly upregulated following injury in iCRT14-treated animals and that showed a significant (false discovery rate [FDR] ≤ 1e-3) reduction in injury-induced expression relative to untreated animals. TCF-inhibited transcripts (purple) were defined as transcripts that were significantly upregulated following injury in iCRT14-treated animals and that showed a significant increase in injury-induced expression relative to untreated animals. (**D**) Average relative chromatin accessibility plot of peaks containing the AP-1 TFBM during head and foot regeneration in both untreated and iCRT14-treated samples. iCRT14 treatment prolonged the injury-induced increase in chromatin accessibility associated with the AP-1 TFBM (i.e., 12-O-tetradecanoylphorbol 13-acetate[TPA] response element [TRE]) during regeneration. (**E, F**) Scatter plots depicting the effect of 5 µM iCRT14 during head regeneration on the injury-induced expression of genes that were specific to head regeneration in untreated animals. TCF inhibition blocks the upregulation of head-specific genes during head regeneration. (**G, H**) Comparison of the average $\log_2$FC in (**G**) transcript abundance or in (**H**) chromatin accessibility between head and foot regenerates treated with 5 µM iCRT14 at 8 hpa. iCRT14 virtually abolished all context-specific transcription at 8 hpa. Features that did not show a significant (FDR ≤ 1e-3) difference in the injury response between head and foot regenerates when compared to 0 hpa controls are highlighted in gray. Features that did show a significant difference are highlighted in red. The dotted line indicates perfect correlation between regeneration types.

The online version of this article includes the following source data and figure supplement(s) for figure 3:

**Source data 1.** Excel workbook containing all tables that were used to generate plots for *Figure 3C–H*.

**Figure supplement 1.** iCRT14 treatment recapitulates head and foot regeneration phenotypes reported by Gufler et al.

**Figure supplement 2.** TCF is required for the downregulation of aspects of the injury response and the onset of structure-specific expression.

**Figure supplement 2—source data 1.** Excel workbook containing tables that were used to generate plots for *Figure 3—figure supplement 2E, F*.

that 5 μM iCRT14 significantly inhibited both head and foot regeneration relative to dimethyl sulfoxide (DMSO)-treated controls (*Figure 3—figure supplement 1A, B*), thus recapitulating previously reported results. We therefore proceeded with generating ATAC-seq and RNA-seq libraries from bisected animals undergoing either head or foot regeneration in the presence of 5 μM iCRT14, repeating the injury conditions used for our untreated sequencing libraries. We collected 0, 8, and 12 hpa timepoints for the RNA-seq timecourse, and 0, 3, 8, and 12 hpa timepoints for the ATAC-seq dataset.

We first determined if our RNA-seq data recapitulated the previously reported finding that TCF inhibition prevents the downregulation of head-specific genes during foot regeneration. Indeed, as reported by *Gufler et al., 2018*, we found that iCRT14 inhibited the downregulation of the head-specific genes *brachyury1* and *β-catenin* in foot regenerating tissue (*Figure 3—figure supplement 1C, D*). We next validated our ATAC-seq data by assessing the effects of iCRT14 on chromatin accessibility near TCF TFBMs to determine if we could detect perturbations in TCF-dependent chromatin remodeling. Our chromVAR analysis revealed that iCRT14 treatment resulted in a diminished and delayed increase in average TCF TFBM accessibility during head regeneration (*Figure 3A*), thus demonstrating our ability to capture changes in chromatin state induced by TCF inhibition.

After validating the data in our iCRT14-treated dataset, we next sought to determine if other head-specific genes beyond those reported by *Gufler et al., 2018* were also inappropriately expressed during foot regeneration when TCF was inhibited. Indeed, we found that several additional head-specific genes—most notably the canonical Wnt signaling components *wnt9/10c*, *wnt3*, and *sp5*—were not appropriately downregulated during foot regeneration by 8 hpa (*Figure 3B*). Our data therefore support the conclusion that TCF is required for the downregulation of a subset of head-specific genes during foot regeneration.

In addition, we found that TCF was also required for the downregulation of transcripts associated with the initial wound response observed during both head and foot regeneration. Several TFs that were transiently upregulated at 3 hpa in untreated animals—including the bZIP-domain containing TFs *creb*, *jun*, and *fos*—showed prolonged upregulation when TCF was inhibited (*Figure 3—figure supplement 2A–C*). In addition, we found that iCRT14 caused an increase in chromatin accessibility near the injury-induced TPA response element (TRE), which is bound by bZIP TFs, in both head and foot regenerates at 8 and 12 hpa (*Figure 3D*). We also observed similar increases for the injury-responsive Egr TFBM (*Figure 3—figure supplement 2D*). These data indicate that TCF is required for the downregulation of early wound-responsive TFs during the transition from the generic wound response to head and foot-specific transcription.

Although Wnt signaling is known to be both necessary and sufficient for head specification in *Hydra*, it is not known which of the transcripts expressed during head regeneration are TCF-dependent. To address this question, we identified head regeneration-specific transcripts that showed reduced upregulation in the presence of iCRT14. We found that TCF was required for the upregulation of numerous head-specific transcripts, including several Wnt signaling components such as *sFRP*, *frizzled4/9/10*, *axin*, and *tcf* itself (*Figure 3E, F*). This potential autoregulatory relationship is consistent with the existence of a Wnt-based feedback loop in *Hydra*, which is thought to provide the molecular basis for the self-maintaining properties of the head organizer (*Gee et al., 2010*; *Nakamura et al., 2011*). Other notable TCF-dependent head-specific transcripts included the TFs *budhead* (*Martinez et al., 1997*) and *prdl-a* (*Gauchat et al., 1998*) and the bmp inhibitor *gremlin-like* (*Watanabe et al., 2014*). The diminished expression of these transcripts highlights potential interactions between Wnt signaling and other regulatory genes implicated in *Hydra* patterning.

We found that TCF inhibition also blocked foot regeneration-specific transcription. Most significantly, all TFs that were enriched in foot regenerating tissue at 12 hpa in untreated animals showed decreased expression in the presence of iCRT14 (*Figure 3C, D*). This suggests a previously unknown role for TCF in the activation of foot-specific transcriptional programs during regeneration and provides a possible molecular basis for the inhibition of foot regeneration by iCRT14 (*Gufler et al., 2018*).

Given the apparent requirement for TCF in the activation of both head and foot-specific transcriptional programs, we next examined how TCF inhibition influenced transcriptional divergence transcriptome-wide. At 8 hpa, we found that iCRT14 blocked virtually all structure-specific transcription, with transcript abundance and chromatin accessibility in head and foot regenerates appearing nearly indistinguishable (*Figure 3G, H*). At 12 hpa, however, we began to see indications of context-

dependent transcription in both our chromatin accessibility and transcriptomic data (*Figure 3—figure supplement 2E, F*). These data therefore demonstrate a global delay in transcriptional divergence and suggest that TCF plays a critical role in the transition from the generic wound response to head and foot-specific transcription during regeneration. This may also indicate that the transient activation of Wnt signaling during early foot regeneration is required to potentiate aboral patterning.

## A conserved injury-responsive gene regulatory network may directly activate Wnt signaling during regeneration

Our observation that iCRT14 inhibited the expression of some Wnt signaling components during regeneration but not others suggested the existence of two distinct regulatory paradigms for injury-induced Wnt component expression: a TCF-dependent mode of activation and a TCF-independent mode of activation. We also noted that the genes exhibiting TCF-independent upregulation were primarily those that were upregulated as part of the early generic injury response, such as *wnt9/10c*, *wnt3*, *wntless*, *sp5*, and *β-catenin*. We therefore hypothesized that these transcripts may be upregulated by injury-responsive TFs, which could account for their expression when TCF was inhibited.

To identify injury-responsive TFs that could plausibly regulate Wnt signaling components during the generic injury response, we performed a systematic candidate screen using our ATAC-seq and RNA-seq datasets. A TF was considered to be a likely candidate if it matched the following criteria when comparing 0 and 3 hpa timepoints: (1) its binding motif was associated with an injury-induced increase in chromatin accessibility, (2) it was transcriptionally upregulated following injury, (3) its binding motif was significantly enriched in peaks that increased in accessibility following injury, and (4) its binding motif was found in predicted cis-regulatory elements near injury-responsive Wnt signaling components. Although these criteria do not definitively eliminate any given TF as a possible regulator, they can provide a basis for evaluating the relative plausibility of different candidates.

Our screen identified ETS, c/EBP, AP-1, and ATF TFBMs as candidate motifs driving injury-induced expression of at least one Wnt signaling component (*Figure 4A*). Among our list of candidates, the ATF TFBM (also known as the cAMP response element [CRE])—which can be bound by the bZIP TFs Jun, Fos, CREB, and ATF (*Eferl and Wagner, 2003*)—best matched our criteria. Of all TFBMs included in our analysis, the CRE TFBM was associated with the largest increase in accessibility from 0 to 3 hpa and showed the strongest enrichment in injury-responsive peaks (*Supplementary file 3*). These chromatin changes were also correlated with the upregulation of several bZIP TFs that are predicted to bind the CRE, including *jun*, *fos,* and two CREB-like TFs (*Figure 4—figure supplement 1A–D*). Furthermore, we identified instances of the CRE in the putative regulatory sequences of *wntless*, *wnt9/10 c*, *wnt3*, and *sp5*. In addition, the CRE-containing peaks near *wntless*, *wnt9/10c*, and *wnt3* increased in accessibility from 0 to 3 hpa during head and foot regeneration, indicating increased activity within those cis-regulatory elements (*Figure 4B–D*). Overall, this analysis indicates that injury-responsive bZIP TFs are the most plausible regulators of canonical Wnt signaling components during the early generic wound response. This potential association between injury-responsive TFs and positive regulators of canonical Wnt signaling provides a mechanistic hypothesis for the link between injury and the reactivation of patterning pathways during *Hydra* regeneration.

## Puncture wounds induce ectopic head formation when pre-existing organizers are absent

Our transcriptomic analysis of head and foot regeneration suggested that canonical Wnt signaling was activated as part of a generic response to injury during early head and foot regeneration, and that the onset of head regeneration-specific Wnt activity did not occur until after the initial generic injury response had ended. We therefore hypothesized that experimentally preventing wound closure could lead to a sustained generic injury response, thus prolonging the generic phase of injury-induced Wnt signaling. Because ectopic Wnt signaling activity is sufficient to induce the formation of ectopic head structures in *Hydra* (*Broun et al., 2005*; *Gee et al., 2010*), we predicted that prolonged injuries could result in ectopic head formation.

To test this prediction, we first established a method for generating a constitutively active injury signal, which we accomplished by transversely impaling *Hydra* on fishing line at the midpoint of their

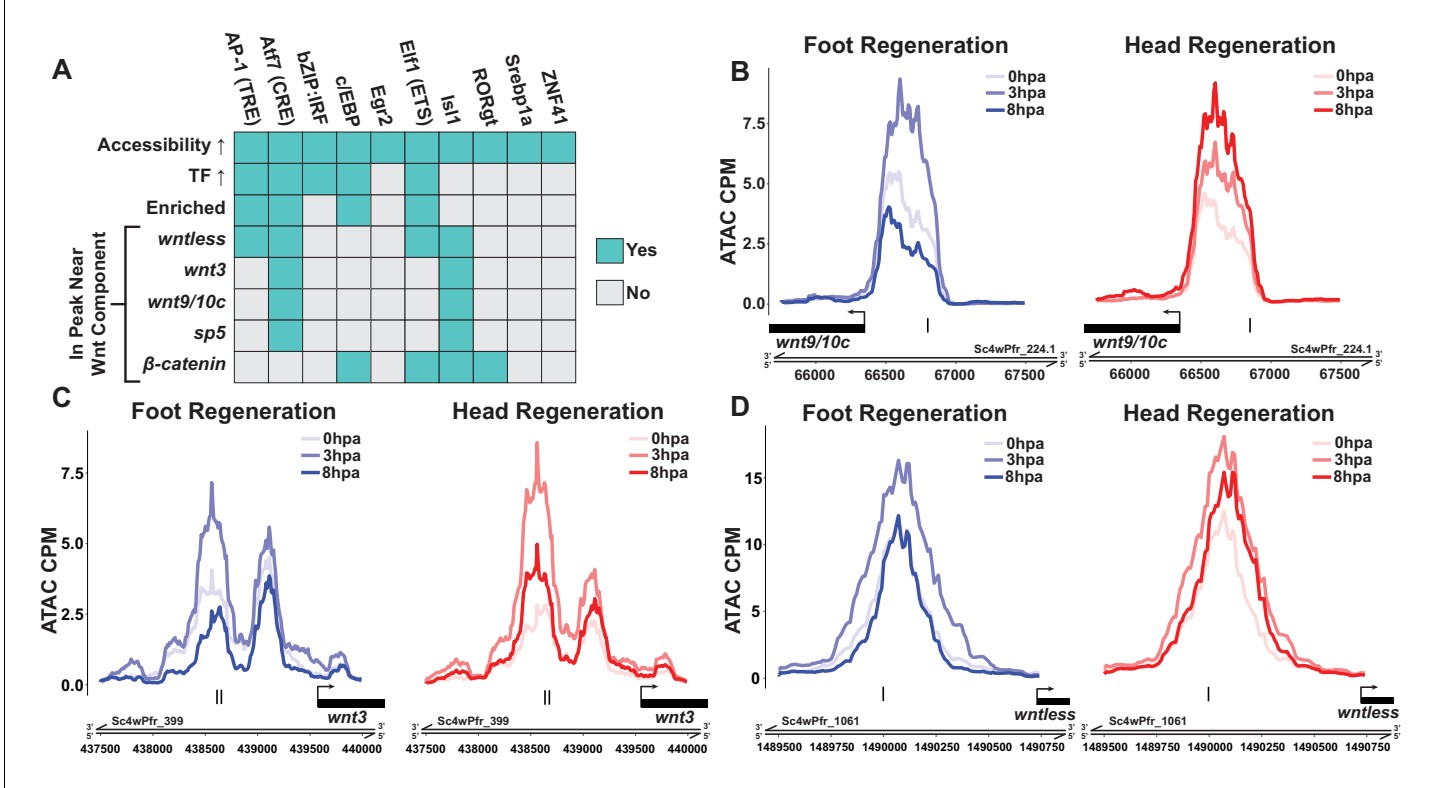

**Figure 4.** Canonical Wnt signaling genes may be directly activated by injury-responsive transcription factors. (**A**) Plot depicting which injury-responsive transcription factor binding motifs (TFBMs) matched the criteria of a plausible regulator of injury-induced Wnt component expression. The AP-1, Atf7, c/EBP, and Elf1 TFBMs are plausible candidate regulators of injury-induced Wnt component expression. 'Accessibility ↑' refers to TFBMs that were associated with a significant increase in chromatin accessibility at 3 hr post amputation (hpa) as calculated by chromVAR. 'TF ↑' refers to TFBMs with at least one transcription factor that could plausibly bind that TFBM showing significant upregulation during head and foot regeneration at 3 hpa. 'Enriched' refers to TFBMs that were significantly enriched in peaks that showed significant increases in accessibility during head and foot regeneration at 3 hpa when compared to peaks that did not increase in accessibility using HOMER. The remaining rows indicate whether or not a given TFBM was found in ATAC-seq peaks located near each respective injury-responsive Wnt signaling component. (**B–D**) ATAC-seq accessibility data for Wnt signaling gene loci during regeneration. The presumptive promoters of injury-induced Wnt signaling genes are likely directly regulated by basic leucine zipper transcription factors. Black tick marks indicate predicted cAMP response element (CRE) hits in putative promoter regions. CRE hits were identified using HOMER. CPM: average ATAC-seq counts per million calculated using a 10 bp bin size.

The online version of this article includes the following figure supplement(s) for figure 4:

**Figure supplement 1.** Basic leucine zipper (bZIP) transcription factors are transiently upregulated during head and foot regeneration at 3 hr post amputation.

oral-aboral axis (*Figure 5A*). This created two large puncture wounds that could not close while the fishing line remained in place. To test the sufficiency of prolonged injury to induce ectopic head organizers, we impaled animals and then either removed the fishing line immediately (0 hr timepoint) or left it in place for 12 hr. Wound outcomes were assessed 4 days post injury. We found that neither the 0 hr nor the 12 hr impalement injuries resulted in aberrant patterning phenotypes (*Figure 5B, C*), indicating that prolonged injuries are not sufficient to induce ectopic organizers in intact polyps.

Extensive research has shown that *Hydra* organizer tissue generates long-range inhibitory signals that prevent ectopic organizers from forming elsewhere in the body, and that amputating organizers transiently increases the capacity of tissue throughout the body column to form secondary axes (*Cohen and MacWilliams, 1975*; *MacWilliams, 1983b*; *MacWilliams and Kafatos, 1968*; *Wilby and Webster, 1970*). This led us to hypothesize that inhibitory signals from pre-existing organizers were preventing ectopic head formation in impaled animals. We therefore modified our approach by amputating head and foot tissue just prior to impalement. In the absence of pre-existing organizers, both 0 and 12 hr impalement injuries resulted in the formation of ectopic oral structures—ranging

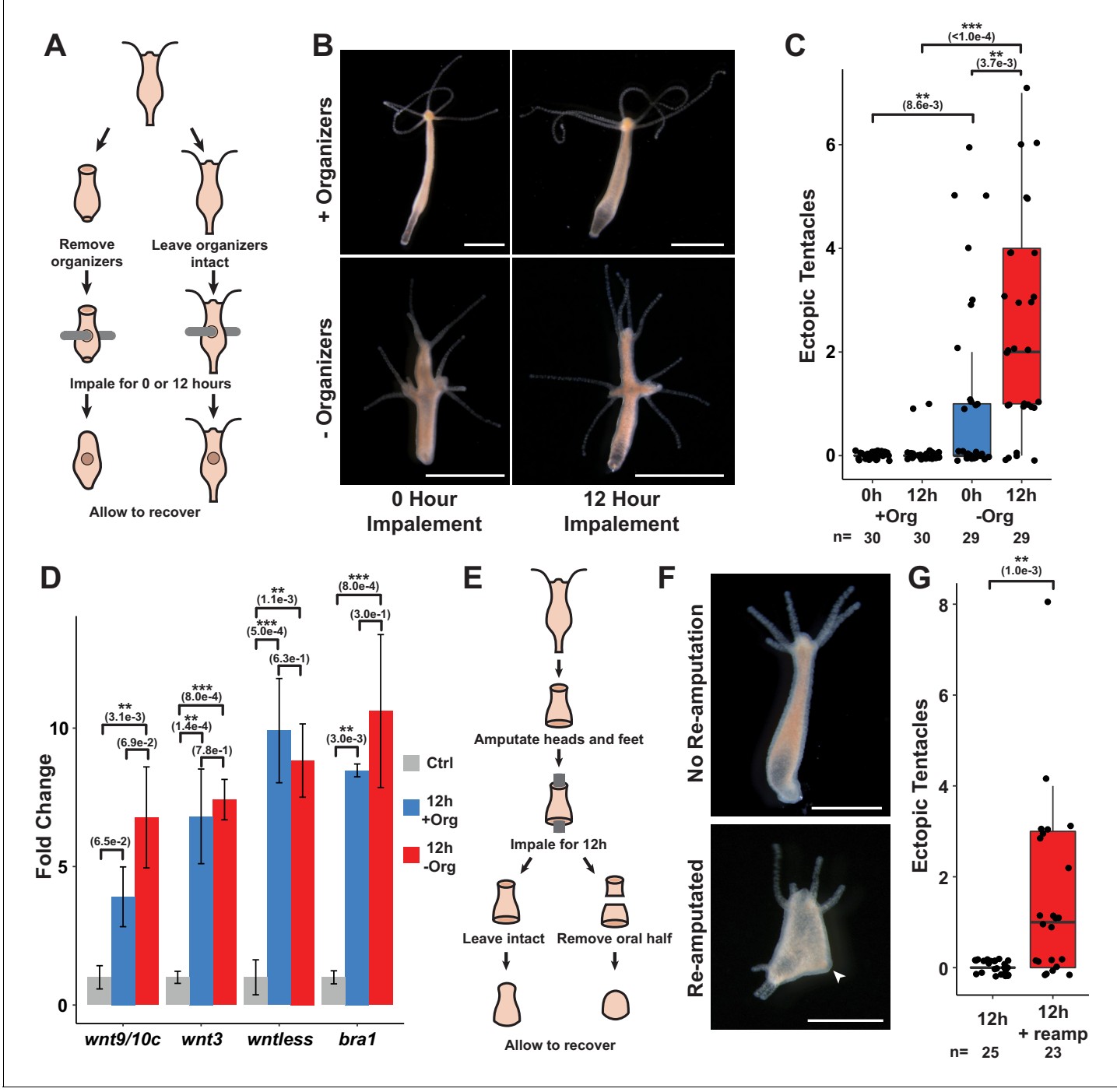

**Figure 5.** Prolonged injuries induce ectopic head formation. (**A**) Experimental design to test for the sufficiency of injury to induce new organizer formation in tissue with or without pre-existing organizers. (**B**) Prolonged injuries induce ectopic head regeneration when pre-existing organizers are removed. Scale bars indicate 1 mm. (**C**) Quantification of the number of ectopic tentacles induced by various impalement injury conditions. Both the presence of pre-existing organizers and the duration of the injury signal significantly contributed to the ectopic head regeneration phenotype. –Org indicates that organizers were removed just prior to impalement, and +Org indicates that organizers were left intact. (**D**) RT-qPCR results for head-specific transcripts in tissue that had been impaled for 12 hr. The Wnt signaling genes *wnt3*, *wntless*, and *wnt9/10c* and the head-specific transcription factor *brachyury1* (*bra1*) are significantly upregulated after 12 hr of impalement. Three biological replicates, each consisting of tissue originating from 15 polyps, were generated for each treatment. Error bars indicate standard deviation. All statistical analyses were performed on $2^{-\Delta Cq}$ values using Tukey's HSD in R. Numbers in parenthesis indicate p-values. (**E**) Experimental design to test both the sufficiency of injury to induce ectopic heads during foot regeneration and the inhibitory capacity of oral-facing amputations. (**F**) Prolonged injuries induce ectopic heads at aboral-facing amputations, but only if prolonged oral-facing amputations are removed. Scale bars indicate 0.5 mm. The white arrowhead indicates a regenerated basal disk. (**G**)

*Figure 5 continued on next page*

*Figure 5 continued*

Quantification of the number of ectopic tentacles induced by various impalement injury conditions. +reamp indicates that the oral half of the regenerating tissue fragment was removed after 12 hr of impalement. HR: head regeneration; FR: foot regeneration, * indicates p-value≤0.05, ** indicates p-value≤0.01, *** indicates p-value≤0.001. All animals were imaged 4 days post injury, giving sufficient time for any removed structures to regenerate.

The online version of this article includes the following figure supplement(s) for figure 5:

**Figure supplement 1.** Puncture injuries induce Wnt pathway gene expression.

from single tentacles to fully formed heads—at one or both of the presumptive impalement sites (*Figure 5B, C*). Importantly, the extent of this phenotype was dependent on the duration of the injury signal as animals impaled for 12 hr grew significantly more ectopic tentacles than animals that had been impaled and then immediately removed from the fishing line (*Figure 5C*). These findings demonstrate that puncture wounds can induce ectopic head organizer formation, and that pre-existing organizers prevent injury-induced ectopic head formation in intact polyps. In addition, the finding that prolonged injuries only resulted in ectopic oral structures and never ectopic aboral structures—despite the fact that *Hydra* body column tissue is equally capable of both oral and aboral regeneration—indicates that oral regeneration is the default outcome promoted by injury.

## Prolonged puncture wounds induce the expression of Wnt pathway components

Because Wnt signaling is both necessary and sufficient for head organizer formation in cnidarians (*Broun et al., 2005*; *Gee et al., 2010*; *Gufler et al., 2018*), we interpret ectopic head formation following impalement as evidence that puncture wounds can induce Wnt signaling. Consistent with this conclusion, we found that treating animals with 5 μM iCRT14 following transverse impalement significantly reduced the number of ectopic tentacles at 4 days post injury, indicating that the phenotype was TCF dependent (*Figure 5—figure supplement 1A*). Furthermore, using RT-qPCR, we found that the Wnt signaling components *wnt9/10c*, *wnt3*, and *wntless* as well as the head-specific TF *brachyury1* were significantly upregulated in tissue that had been transversely impaled for 12 hr (*Figure 5D*).

While performing transverse impalement experiments, we noted that ectopic head formation was strain-dependent as the Basel and 105 strains formed ectopic heads, but the AEP strain did not (*Figure 5—figure supplement 1B–E*). We nonetheless found that Wnt signaling components were significantly upregulated following 12 hr of impalement in all three strains (*Figure 5D*, *Figure 5—figure supplement 1F, G*). We therefore speculate that the differences in injury outcomes among these strains may be caused by genetic variation affecting inhibition levels in body column tissue.

Despite the dramatic effects that pre-existing organizers had on patterning outcomes following impalement, we found that the presence or absence of head and foot tissue did not significantly affect the expression of Wnt signaling components in impaled tissue after wound closure was blocked for 12 hr (*Figure 5D*). However, the ectopic heads induced by impalement expressed Wnt pathway-associated transcripts at significantly higher levels than impaled tissue that did not form ectopic heads (*Figure 5—figure supplement 1H*). Thus, the presence of pre-existing organizers inhibited the expression of injury-induced Wnt pathway genes, but it remains unclear when and how this inhibition occurred.

## Regenerating heads can inhibit ectopic head formation at aboral-facing amputations

The role that pre-existing head and foot tissue played in preventing ectopic head formation following transverse impalement raised the question of why ectopic heads do not form at aboral-facing injuries when both the head and foot are amputated. We hypothesized that in such cases heads do not form at aboral-facing amputations because—although head patterning pathways are initially activated in both injuries—the oral-facing amputation recovers its ability to inhibit ectopic heads faster than the aboral-facing amputation, thus preventing ectopic head formation at the aboral pole. This hypothesis is consistent with the observations that regenerating heads gradually regain their inhibitory capacity over time and that head regeneration is faster when head amputations are

located closer to the oral pole (*MacWilliams, 1983a*). We therefore sought to test if regenerating head tissue can prevent ectopic head formation at aboral-facing amputations.

To address this question, we first sought to increase the capacity for aboral-facing amputations to form ectopic heads by amputating both the head and aboral third of *Hydra* polyps and impaling the regenerating animals through the two resulting amputation injuries for 12 hr (*Figure 5E*). After impalement, we either let the animals recover unperturbed or we re-amputated the regenerating head promptly after the animals were removed from the fishing line. Thus, the former set of injury conditions prolonged both the oral and the aboral-facing amputations, while the latter set of conditions only prolonged the aboral-facing amputation. We found that when both amputations were prolonged there was no ectopic head formation (*Figure 5F, G*). However, when only the aboral-facing amputation was prolonged, head structures formed at both poles of the oral-aboral axis in ~61% of cases (13/23 animals). These findings demonstrate that prolonged aboral-facing amputation injuries can induce head formation, and that regenerating heads can inhibit ectopic head formation at aboral-facing amputations. Notably, foot regeneration occurred in all cases, such that one pole of the regenerated animal had a head while the other pole had both a foot and a full or partial head (*Figure 5F*). This therefore suggests that the specification of foot tissue during regeneration occurs independently from head specification.

## Discussion

Using RNA-seq and ATAC-seq, we have comprehensively characterized transcription during the first 12 hr of head and foot regeneration in *Hydra*. We found that the initial response to amputation at 3 hpa did not depend on the structure being regenerated and included widespread apoptosis and the upregulation of canonical Wnt signaling components at both oral- and aboral-facing wounds. During this initial response, we also observed symmetric increases in chromatin accessibility near predicted TCF binding sites, suggesting that Wnt signaling is activated during early head and foot regeneration. By 8 hpa, however, Wnt signaling component expression and chromatin accessibility near TCF TFBMs became specific to head regeneration. Inhibiting TCF delayed the onset of head and foot regeneration-specific transcription transcriptome-wide, demonstrating a central role for the Wnt-responsive TF during the two types of regeneration. We found that Wnt pathway components were also upregulated in non-regenerative puncture wounds, supporting the conclusion that these genes are part of the generic transcriptional response to injury. Furthermore, removing pre-existing organizers induced puncture wounds to form ectopic head organizers, and experimentally blocking wound closure for 12 hr increased the penetrance of this phenotype. This demonstrates that injury signals are capable of initiating head patterning, even in the context of non-amputation injuries. We also found that prolonged foot amputation injuries could induce ectopic head formation, and that this phenotype could be suppressed by regenerating head tissue. To our knowledge, our work is the first demonstration in any animal that injuries alone are sufficient to induce secondary axes.

Based on our findings, we propose the following model of Wnt signaling activity during *Hydra* regeneration: injuries activate canonical Wnt signaling regardless of the injury type, which has the potential to give rise to a new head organizer through an autocatalytic feed-forward loop; however, in the case of foot regeneration or non-amputation injuries, inhibitory signals from pre-existing head tissue or oral-facing amputations block the Wnt feed-forward loop and thus prevent ectopic head formation (*Figure 6*). A version of this model was originally proposed over 30 years ago based on an extensive body of research that used classical tissue manipulation techniques (*Gierer and Meinhardt, 1972*; *Kobatake and Sugiyama, 1989*; *MacWilliams, 1983a*; *Newman, 1974*); however, this original model lacked an underlying molecular mechanism, relying instead on abstract theoretical actors to account for the injury-induced activation of head patterning—a phenomenon referred to as the 'injury effect' (*Kobatake and Sugiyama, 1989*; *MacWilliams, 1983a*). Our work suggests that the injury effect documented in previous research is the result of injury-induced Wnt signaling, thus providing an updated molecular framework that is highly consistent with the pre-existing theoretical model.

An alternative model of oral patterning during *Hydra* regeneration proposed that canonical Wnt signaling is rapidly activated within 1.5 hpa exclusively at oral-facing amputation injuries as a result of head regeneration-specific apoptosis (*Chera et al., 2009*). However, we found no evidence of these early asymmetries. Instead, we found that the apoptotic and transcriptional response to

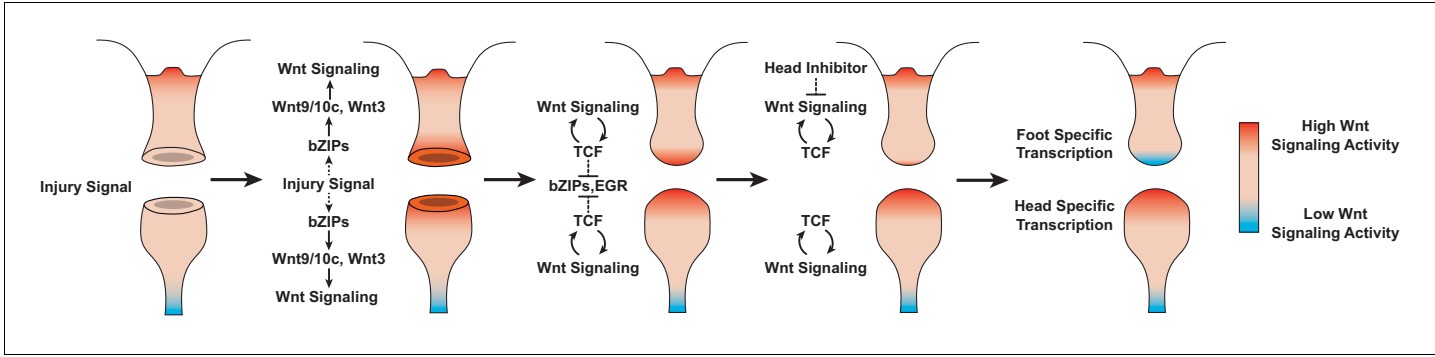

**Figure 6.** Proposed model of canonical Wnt signaling activity following mid-gastric bisection in *Hydra*. Injury triggers a Wnt signaling cascade regardless of tissue context through the direct upregulation of the Wnt ligands *wnt9/10c* and *wnt3* via injury-responsive basic leucine zipper transcription factors. An increase in Wnt signaling then represses injury-responsive transcription factors in a TCF-dependent manner. While Wnt signaling has the capacity for autocatalytic amplification, this only occurs during head regeneration as the autocatalytic amplification is blocked during foot regeneration by an uncharacterized inhibitor originating from pre-existing oral tissue. HR: head regeneration; FR: foot regeneration. Red coloration denotes high Wnt signaling activity. Blue coloration denotes low Wnt signaling activity.

The online version of this article includes the following figure supplement(s) for figure 6:

**Figure supplement 1.** *notum* is expressed in head tissue in whole uninjured *Hydra*.

amputation was indistinguishable between head and foot regeneration at 3 hpa. The recently reported finding that MAP kinases—highly conserved upstream regulators of wound repair and apoptosis—showed identical levels of activation during head and foot regeneration is also consistent with the existence of an initial generic response to injury (*Tursch et al., 2020*). Finally, our finding that TCF is required for the activation of foot-specific transcription during regeneration appears to be inconsistent with injury-induced Wnt signaling only occurring during head regeneration.

Our results highlight the existence of notable similarities in the transcriptional regulation of Wnt signaling components across distantly related regeneration models. In addition to *Hydra*, Wnt signaling components are upregulated as part of an early generic response to injury in the cnidarian species *Nematostella vectensis* and *Clytia hemisphaerica* as well as in planarians (*DuBuc et al., 2014*; *Gurley et al., 2010*; *Petersen and Reddien, 2009*; *Schaffer et al., 2016*; *Sinigaglia et al., 2020*; *Stückemann et al., 2017*; *Wurtzel et al., 2015*). In addition, recent research has demonstrated that MAP kinase (MAPK) signaling is required for the upregulation of Wnt pathway components during the generic wound response in both planarians and *Hydra* (*Owlarn et al., 2017*; *Tursch et al., 2020*). Although the TFs that regulate Wnt components during regeneration in planarians are unknown, our systematic analysis of injury-responsive Wnt component regulatory sequences in *Hydra* found that all of the most plausible regulators of Wnt pathway genes during *Hydra* regeneration are TFs that are directly regulated by MAPK signaling (*Whitmarsh, 2007*). This raises the possibility that MAPK-induced TFs are conserved regulators of Wnt pathway transcripts during regeneration. Both *Nematostella* and bilaterians diverged from *Hydra* over 500 million years ago (*Erwin et al., 2011*; *Khalturin et al., 2019*). Thus, these similarities span sizable evolutionary timescales and suggest the existence of deeply conserved aspects of the gene regulatory networks involved in the response to injury during regeneration.

There nonetheless exist important differences in the regulation of Wnt signaling during regeneration in bilaterians and *Hydra*. In planarians, Wnt signaling is restricted to tail regeneration through the head regeneration-specific upregulation of the Wnt inhibitor *notum* (*Petersen and Reddien, 2011*). In cnidarians, *notum* and canonical Wnt ligands are co-expressed in both uninjured and regenerating head tissue (*Petersen et al., 2015*; *Siebert et al., 2019*, *Figure 6—figure supplement 1*). Furthermore, in contrast to *Hydra*, pre-existing organizers do not appear to influence patterning during planarian regeneration (*Petersen and Reddien, 2011*). In acoels, Wnt pathway components are not part of the generic transcriptional response to injury and are instead rapidly upregulated specifically at aboral-facing amputations (*Ramirez et al., 2020*). In addition, the TF Egr is responsible for the injury-induced expression of *wnt-3* in acoels, but Egr does not appear to be a plausible

regulator of injury-responsive Wnt pathway genes in *Hydra*. Thus, there are likely significant differences in the regulation of Wnt signaling during regeneration in *Hydra*, planarians, and acoels.

There also appear to be differences in the downstream effects of Wnt pathway gene upregulation during whole-body regeneration in *Hydra* and bilaterians. In planarians, there are conflicting reports on whether canonical Wnt signaling is activated as part of the generic response to injury (*Stückemann et al., 2017*; *Sureda-Gómez et al., 2016*); however, knocking down Wnt signaling genes causes head regeneration to occur at both anterior and posterior-facing amputations (*Gurley et al., 2008*; *Petersen and Reddien, 2008*), demonstrating that the upregulation of Wnt transcripts is not required for the specification of head tissue. Similar knockdown phenotypes have also been reported in acoels (*Srivastava et al., 2014*). This is in contrast to *Hydra*, where TCF inhibition blocks regeneration of both the oral and aboral ends. Our findings that TCF TFBM accessibility increased during both early head and foot regeneration and that non-amputation injuries could induce ectopic heads strongly suggest that the transcriptional upregulation of Wnt signaling components during the generic wound response is accompanied by the activation of Wnt signaling in *Hydra*. Furthermore, the finding that inhibiting TCF blocks head and foot regeneration indicates that this activation likely plays an important role in both types of regeneration (*Gufler et al., 2018*). However, further characterization of the Wnt pathway in *Hydra*, in particular the identification of direct Wnt pathway targets and the characterization of β-catenin protein dynamics during the early wound response, will be required to fully understand the role of Wnt signaling during *Hydra* regeneration.

# Materials and methods

**Key resources table**

| Reagent type (species) or resource | Designation | Source or reference | Identifiers | Additional information |
|---|---|---|---|---|
| Strain, strain background (*Hydra vulgaris*) | AEP | *Martin et al., 1997*; *Wittlieb et al., 2006* | | |
| Strain, strain background (*Hydra vulgaris*) | 105 | *Chapman et al., 2010* | | |
| Strain, strain background (*Hydra vulgaris*) | Basel | *Technau and Holstein, 1995* | | |
| Commercial assay or kit | Tagment DNA Enzyme and Buffer Small Kit | Illumina | 20034197 | |
| Commercial assay or kit | Kapa mRNA-seq Hyper kit | Roche | KK8581 | |
| Commercial assay or kit | MinElute PCR Purification Kit | QIAGEN | 28004 | |
| Commercial assay or kit | DNase Set | QIAGEN | 79254 | |
| Commercial assay or kit | RNA Clean and Concentrator kit | Zymogen | R1017 | |
| Chemical compound, drug | iCRT14 | Sigma-Aldrich | SML0203 | |
| Software, algorithm | R | R | RRID:SCR_001905 | |
| Other | Agencourt AMPure XP beads | Beckman Coulter | A63881 | |
| Other | 2X NEBNext master mix | NEB | M0541S | |
| Other | M-MLV RNase H Minus Point Mutant Reverse Transcriptase | Promega | M3682 | |
| Other | SsoAdvanced universal SYBR green master mix | Bio-Rad | 1725271 | |

## Code and data availability

All code used in this study is available both as a git repository at https://github.com/cejuliano/jca-zet_regeneration_patterning (*Cazet and Juliano, 2021*; copy archived at swh:1:rev: 6e9b9067f0adf1bd223c5ee0194d75847fa18321) and on Dryad at https://doi.org/10.25338/B8S612. FASTQ files of raw ATAC-seq and RNA-seq reads, expression matrices for ATAC-seq and RNA-seq reads mapped to the *Hydra* 2.0 genome reference, consensus peak files, and bigwig genome tracks of individual and pooled ATAC-seq replicates are available through the Gene Expression Omnibus

under the accession GSE152994. The *Hydra* 2.0 genome gene model IDs associated with the gene names used throughout this study are provided in *Table 3*.

## ATAC-seq library preparation

For each ATAC-seq replicate, ~15 whole, bud-free *H. vulgaris* (strain 105) polyps that had been fed once weekly were starved for 2 days and then transversely bisected at the midpoint of their oral-aboral axis. Regeneration was then allowed to proceed for 0, 3, 8, or 12 hr. Regenerating tips corresponding to ~1/3 of the total regenerate length were then isolated from head and foot regenerates and used for generating ATAC-seq libraries. For iCRT14-treated samples, 10 mM iCRT14 (SML0203; Sigma-Aldrich, St. Louis, MO) dissolved in dimethyl sulfoxide (DMSO) was diluted in *Hydra* medium to a final concentration of 5 µM. iCRT14 solution was added to the media 2 hr before amputation and was left in place until the tissue was collected for library preparation. For each treatment, 3–5 biological replicates were prepared (replicates are listed in *Table 1*). The number of replicates was chosen based on the widely used standard of three biological replicates per treatment for high-throughput sequencing experiments, although we slightly increased the number of replicates to increase the statistical power of differential tests.

To generate ATAC-seq libraries, we made use of a modified version of the OMNI-ATAC protocol as described previously (*Corces et al., 2017*; *Siebert et al., 2019*). Briefly, regenerating tips were washed once with 1 ml of chilled *Hydra* dissociation medium (DM) (*Gierer et al., 1972*) and then homogenized in 1 ml fresh DM using ~30 strokes of a tight-fitting dounce. Cells were then pelleted at 500 RCF for 5 min at 4°C in a benchtop centrifuge and subsequently lysed in 50 µl of chilled resuspension buffer (RSB; 10 mM Tris-HCl, pH 7.4, 10 mM NaCl, 3 mM MgCl$_2$) with 0.1% Tween-20, 0.1% NP-40, and 0.01% digitonin for 3 min on ice. Lysis was halted by adding 1 ml of RSB with 0.1% Tween-20. The lysate's nuclear concentration was then quantified by loading 19 µl of lysate and 1 µl of 0.2 mg/ml Hoechst 33342 onto a Fuchs-Rosenthal hemocytometer. A volume of lysate corresponding to ~50,000,000 nuclei was then aliquoted and spun at 500 RCF for 10 min at 4°C in a benchtop centrifuge. The resulting crude nuclear pellet was then resuspended in 50 µl of tagmentation solution (1× TD buffer [20034197; Illumina, San Diego, CA], 33% phosphate-buffered saline [PBS], 0.01% digitonin, 0.1% Tween-20, 5 µl TDE1 [Illumina 20034197]) and shaken at 1000 rpm for 30 min at 37°C. Tagmentation was halted by adding 250 µl of PB buffer from a QIAGEN MinElute PCR Purification Kit (28004; QIAGEN, Hilden, Germany).

Tagmented DNA was purified using a QIAGEN MinElute PCR Purification Kit following the standard manufacturer's protocol. Libraries were then amplified with 2X NEBNext master mix (M0541S; NEB, Ipswitch, MA) using cycle numbers determined by quantitative reverse transcription PCR (qPCR) as described in the standard ATAC-seq protocol (*Buenrostro et al., 2015*; *Buenrostro et al., 2013*). Agencourt AMPure XP beads (A63881; Beckman Coulter, Pasadena, CA) were then used to purify libraries and restrict fragment sizes to between 100 and 700 base pairs. Libraries were then pooled and sequenced on an Illumina HiSeq4000 using 2 × 150 bp reads.

## ATAC-seq data processing

Sequencing adapters, stretches of low-quality base calls, and unpaired reads were removed from the raw sequencing data using Trimmomatic (*Bolger et al., 2014*). Filtered reads were then mapped to the *Hydra vulgaris* 2.0 genome (arusha.nhgri.nih.gov/hydra/) using Bowtie2 (*Langmead and Salzberg, 2012*). Mitochondrial reads were removed by independently mapping filtered reads to the *Hydra* mitochondrial genome (*Voigt et al., 2008*) and removing mitochondrial reads from the genome-mapped data using Picard Tools (broadinstitute.github.io/picard). Ambiguously mapped reads (defined as having a mapping quality (MAPQ) value of ≤3) and discordantly mapped read pairs were removed using SAMtools (*Li et al., 2009*). PCR duplicates were labeled using Picard Tools and removed using SAMtools.

Peak calling was performed using a modified version of the ENCODE consortium's ATAC-seq analysis pipeline (encodeproject.org/atac-seq) (*Landt et al., 2012*). First, unambiguously mapped deduplicated non-mitochondrial reads were centered over the transposase binding site by shifting + strand reads +4 bp and – strand reads −5 bp using deepTools2 (*Ramírez et al., 2016*). Peaks were then called using Macs2 (*Zhang et al., 2008*) with a permissive p-value cutoff of 0.1. We then generated consensus lists of biologically reproducible peaks using the irreproducible discovery rate (IDR)

framework (*Li et al., 2011*) by identifying peaks that were reproducible (IDR score ≤0.1) across at least three pairwise comparisons of biological replicates in at least one treatment group.

To assess the quality and reproducibility of our ATAC-seq data, we made use of several metrics used by the ENCODE consortium. Because core promoters are expected to be highly accessible, ATAC-seq reads should be strongly enriched near transcription start sites (TSS). The current *Hydra* genome gene models lack UTR annotations, so we determined the TSS enrichment score for each biological replicate by measuring read enrichment near the first start codon of the 2000 most highly expressed genes in the *Hydra* single-cell RNA-seq atlas relative to the average read density ±1 kb from the start codon. We observed between a five- and eightfold TSS enrichment across the individual samples in our dataset. We also calculated the self-consistency and rescue ratios for each treatment group to evaluate consistency across replicates. We found that all treatment groups had self-consistency and rescue ratios less than 2, indicating good overall reproducibility across replicates. A full table of ATAC-seq library metrics is provided in *Table 1*.

## Differential accessibility and chromVAR analysis

Read counts for peaks in our consensus peakset were calculated using the R Diffbind package (*Ross-Innes et al., 2012*). edgeR was then used for downstream differential accessibility analyses (*Robinson et al., 2010*). Peaks that did not have at least 10 counts per million in at least three replicates were excluded from further analyses. Differentially accessible peaks were then identified using quasi-likelihood tests of count data fitted to a negative binomial generalized log-linear model with a fFalse discovery rate (FDR) cutoff of 1e-4. The full results tables for all ATAC-seq pairwise comparisons performed as part of the edgeR analysis are included in *Supplementary file 1*.

For the chromVAR TFBM accessibility analyses, the width of all peaks that showed a significant change during regeneration was fixed at 250 bp and new read counts were generated. Changes in accessibility were then calculated for all TFBMs provided in the custom list of HOMER motifs found in the chromVARmotifs package. Because our unfiltered enrichment results often included redundant TFBMs with highly similar sequence composition and accessibility dynamics, TFBM redundancy was reduced by performing hierarchical clustering on a matrix of pairwise TFBM similarity scores generated by the HOMER compareMotifs function. The TFBM within a cluster that showed the greatest change in accessibility across treatments was then used as the representative motif for that cluster. To identify motifs enriched in injury-responsive peaks at 3 hpa, peaks that showed a significant increase in accessibility during either head or foot regeneration at 3 hpa were compared to all other peaks in our consensus peakset using the HOMER findMotifsGenome function.

## RNA-seq library preparation

30 whole, bud-free *H. vulgaris* (strain 105) polyps that had been fed once weekly were starved for 2 days and then transversely bisected at the midpoint of their oral-aboral axis in batches of 10 to reduce variability in amputation and collection times. iCRT14 treatment conditions followed the same protocol as was described above for ATAC-seq library preparation. Regeneration was then allowed to proceed for 0, 3, 8, or 12 hr. Regenerating tips corresponding to ~1/3 of the total regenerate length were then isolated from head and foot regenerates and frozen in Trizol (15596018; Thermo Fisher Scientific, Waltham, MA) at −80°C. The three individual batches of 10 regenerating tips were then pooled into a single biological replicate, and RNA was subsequently extracted using a standard Trizol RNA purification protocol. For each treatment, three biological replicates of 30 animals each were prepared (replicates are listed in *Table 2*). The number of replicates was chosen based on the widely used standard of three biological replicates per treatment for high-throughput sequencing experiments. DNA contamination was then removed using the QIAGEN DNase Set (79254; QIAGEN) following the manufacturer's protocol. A final cleanup was then performed using a Zymogen RNA Clean and Concentrator kit (R1017; Zymogen, Irvine, CA) following the standard manufacturer's protocol. Strand-specific polyA-enriched libraries were prepared using the Kapa mRNA-seq Hyper kit (KK8581; Kapa Biosystems, Cape Town, South Africa). Untreated samples were sequenced on an Illumina HiSeq4000 with 1 × 100 bp reads. 12 hpa and all iCRT14-treated samples were sequenced on an Illumina NovaSeq using 2 × 150 bp reads.

## RNA-seq data processing and differential gene expression analysis

Sequencing adapters and stretches of low-quality base calls were removed from the raw sequencing data using Trimmomatic. Filtered reads were then mapped to the *Hydra* 2.0 gene models, and read counts per gene were calculated using RSEM (*Li and Dewey, 2011*). A full table of RNA-seq library metrics is provided in *Table 2*. Using edgeR, read counts were normalized and genes that did not have at least two counts per million in at least three replicates were excluded from further analyses. Differentially expressed genes were then identified using quasi-likelihood tests of count data fitted to a negative binomial generalized log-linear model with a FDR cutoff of 1e-3. The full results tables for all RNA-seq pairwise comparisons performed as part of the edgeR analysis are included in *Supplementary file 2*.

To identify genes that were enriched in head or foot tissue in homeostatic *Hydra*, single-cell read counts for epithelial head and foot cells from a previously published *Hydra* single-cell RNA-seq atlas (*Siebert et al., 2019*) were compared using a Wilcoxon rank-sum test. Enrichment was determined using an adjusted p-value cutoff of 1e-6.

To systematically identify and characterize the expression of canonical Wnt signaling genes during regeneration, KEGG-annotated Wnt pathway components were isolated from the *Nematostella vectensis*, *Exaiptasia pallida*, *Hydra vulgaris* 1.0, and *Homo sapiens* genomes. We queried multiple species because many of the cnidarian references used to generate KEGG annotations are highly fragmented and likely incomplete. Putative orthologs of these KEGG references were then identified in the *Hydra vulgaris* 2.0 genome reference using reciprocal blast searches. BLAST hits were then manually evaluated to exclude false positives by examining protein domain composition as determined by InterPro (*Blum et al., 2021*) and performing additional BLAST searches against the NCBI NR reference. Wnt pathway components that had been previously characterized in *Hydra* and were not recovered using KEGG annotations (*naked cuticle*, *wntless*, and *sp5*) were added to the candidate list manually. Additionally, Wnt signaling genes that were not directly associated with the canonical pathway (e.g., non-canonical Wnt signaling) were excluded.

## Vital staining of apoptotic cells using acridine orange

Staining of apoptotic cells using acridine orange was performed using a modified version of a previously described protocol (*Cikala et al., 1999*). Briefly, animals were collected in batches of 15 and incubated in 1 ml 1.6 µM acridine orange diluted in *Hydra* medium for 2 min in the dark. The *Hydra* were then quickly washed twice with 1 ml *Hydra* medium and immediately documented. Staining was performed on whole uninjured *Hydra* and on head or foot regenerating animals following midgastric bisection at 1 or 3 hpa.

## Validation of iCRT14 treatment conditions

To validate the efficacy of iCRT14 in inhibiting head regeneration and foot regeneration, three biological replicates consisting of 30 polyps each were pre-treated for 2 hr in either 5 µM iCRT14 or 0.05% DMSO and then bisected at the midpoint of their oral aboral axis. The animals were then left to recover in either 5 µM iCRT14 or 0.05% DMSO that was regularly refreshed every 12 hr. Head regeneration was assessed by quantifying the number of regenerated tentacles at 60 hpa. Foot regeneration was assessed at 36 hpa by performing a foot peroxidase staining assay as previously described (*Hoffmeister and Schaller, 1985*). Briefly, *Hydra* were relaxed in 2% urethane in *Hydra* medium for 1 min and then fixed for 1 hr at room temperature in 4% paraformaldehyde in *Hydra* medium. Fixative was removed using three quick PBS washes. The samples were then incubated in 5% sucrose in PBS for 24 hr at 4°C. All subsequent steps were performed at room temperature. Excess sucrose was removed using three 5 min washes in PBT (0.1% Tween in PBS). Basal disks were then stained by incubating *Hydra* in a solution of 0.02% diaminobenzidine and 0.003% $H_2O_2$ in PBT for 15 min. Excess stain was removed using three 5 min washes in PBT, and the number of stained disks per biological replicate was quantified. Student's t-test function in R was used to test for significant differences in the number of regenerated tentacles and the proportion of animals with regenerated basal disks.

## *Hydra* tissue manipulation experiments

For transverse impalement experiments, *H. vulgaris* (either Basel, 105, or AEP strain) polyps that had been fed twice weekly were starved for at least 24 hr and then impaled at the midpoint of their oral-aboral axis in batches of ~15 on a single piece of 0.3 mm diameter fishing line. Each treatment had a total of ~30 samples. The sample size was chosen to maximize statistical power while minimizing temporal variability in injury duration. To remove pre-existing organizer tissue before impalement, heads and feet were removed by performing transverse amputations just below the tentacle ring and just above the peduncle, respectively. The *Hydra* were then either removed from the fishing line immediately after being impaled for the 0 hr timepoint or were left on the fishing line for 12 hr and then removed for the 12 hr timepoint. Phenotypes were then documented after 4 days of recovery post impalement. Occasionally, during the 12 hr impalement experiments, the epithelia would heal around the fishing line, thus prematurely terminating the injury signal. Animals from the 12 hr timepoint that underwent premature wound healing were excluded from subsequent analyses. Significant differences in the number of ectopic tentacles at 4 days post injury were identified using an ANOVA followed by Tukey's HSD test in R using the agricolae package.

To test the role of TCF in ectopic head formation induced by transverse impalement, the heads and feet of Basel strain *Hydra* were first removed as described above. The animals were then transversely impaled for 12 hr and subsequently allowed to recover in either 0.05% DMSO or 5 µM iCRT14 in *Hydra* medium that was refreshed every 12 hr for a total of 4 days. Each treatment group consisted of ~30 animals. The number of ectopic tentacles was quantified at 4 days post injury, and the treatment groups were compared using the R Student's t-test function.

For prolonged aboral amputation experiments, transverse cuts were used to remove the head and aboral-most third of Basel strain *H. vulgaris* polyps in batches of ~10–15 animals each. Animals were then impaled through the two resulting amputation wounds using 0.3 mm fishing line. After 12 hr of impalement, *Hydra* were either left to recover undisturbed or their oral-most half was amputated and discarded. To track the original orientation of the oral-aboral axis of impaled tissue fragments, one end of the fishing line was marked with a permanent marker and all *Hydra* were impaled such that their aboral pole pointed towards the marked end of the fishing line. The number of ectopic tentacles was quantified 4 days post injury, and significant differences in the number of ectopic tentacles were determined using Student's t-test in R.

## Quantitative reverse transcription PCR

Transversely impaled Basel, AEP, or 105 strain *Hydra* were prepared as described above, with 15 polyps per biological replicate and a total of three biological replicates per treatment. The number of replicates was chosen based on the widely used standard of three biological replicates per treatment for qPCR experiments. For samples collected immediately after impalement, tissue surrounding the impalement site, corresponding to ~1/4 of the total polyp length, was collected for downstream RNA isolation.

To quantify gene expression in ectopic heads, Basel strain animals lacking pre-existing organizers were impaled for 12 hr and then left to recover. As a control, body column tissue was collected from animals that were impaled for 12 hr with their organizers intact after 4 days of recovery. Both the ectopic heads and the control tissue were collected 4 days post injury.

RNA purification was performed using the same protocol as described above for RNA-seq library preparation. cDNA was synthesized using 1 µg of purified RNA and Promega M-MLV RNase H Minus Point Mutant Reverse Transcriptase (M3682; Promega, Madison, WI) using the manufacturer's recommended protocol for oligo dT-primed synthesis. cDNA was then diluted 1:3 in nuclease-free water for use in qPCR experiments.

Three technical replicates of 10 µl qPCR reactions per sample were prepared using Bio-Rad SsoAdvanced universal SYBR green master mix (1725271; Bio-Rad, Hercules, CA) and were run on a CFX96 Touch Real-Time PCR Detection System (1855195; Bio-Rad). Cq values from technical replicates were pooled for subsequent analyses. *rp49* was used as an internal control to calculate ΔCq values after first being found to give similar results across all treatments when compared to a second housekeeping gene, *actin*. Statistically significant differences in mRNA expression were calculated by performing an ANOVA on $2^{-\Delta Cq}$ values followed by Tukey's HSD test in R using the agricolae package.

## Acknowledgements

We thank Bruce Draper, Gary Wessel, Rob Steele, Bryan Teefy, Sergio Campos, Ben Cox, Thomas Holstein, Anja Tursch, Prashanth Rangan, Jeffrey Farrell, and Stefan Materna for critical reading of the manuscript; the members of the Juliano lab for their thoughtful input on this study; Vanessa Rashbrook, Emily Kumimoto, Siranoosh Ashtari, and Lutz Froenicke from the DNA Technologies and Expression Analysis Core at the UC Davis Genome Center (supported by NIH Shared Instrumentation Grant 1S10OD010786-01) for technical advice and assistance with library preparation and sequencing; and the Vincent J. Coates Genomics Sequencing Laboratory at UC Berkeley for additional library preparation and sequencing.

## Additional information

### Funding

| Funder | Grant reference number | Author |
|---|---|---|
| National Institutes of Health | R35 GM133689 | Celina Juliano |

The funders had no role in study design, data collection and interpretation, or the decision to submit the work for publication.

### Author contributions

Jack F Cazet, Conceptualization, Data curation, Formal analysis, Investigation, Visualization, Methodology, Writing - original draft; Adrienne Cho, Investigation; Celina E Juliano, Conceptualization, Supervision, Funding acquisition, Project administration, Writing - review and editing

### Author ORCIDs

Jack F Cazet  https://orcid.org/0000-0002-7331-5631
Adrienne Cho  https://orcid.org/0000-0001-6837-420X
Celina E Juliano  https://orcid.org/0000-0003-4222-0987

### Decision letter and Author response

Decision letter https://doi.org/10.7554/eLife.60562.sa1
Author response https://doi.org/10.7554/eLife.60562.sa2

## Additional files

### Supplementary files

• Supplementary file 1. Full results tables for all pairwise comparisons performed on ATAC-seq data. Excel workbook containing the results tables generated by edgeR as individual worksheets. Peaks were included in the analysis if they had at least 10 mapped counts per million in at least three samples. The first sheet in the workbook ('Contrasts') provides the formulae associated with the names assigned to each comparison in the workbook. The peakset files ('untreated_consensus_diffbind_labels.bed' and 'full_consensus_diffbind_labels.bed') used to assign the peak ID numbers used in this worksheet are provided in the repositories (Github or Dryad) associated with this study.

• Supplementary file 2. Full results tables for all pairwise comparisons performed on RNA-seq data. Excel workbook containing the results tables generated by edgeR as individual worksheets. The first sheet in the workbook ('Contrasts') provides the formulae associated with the names assigned to each comparison in the workbook. Genes were included in the analysis if they had at least two mapped counts per million in at least three samples. The first sheet in the worksheet ('Contrasts') provides the formulae/descriptions associated with the names given to each comparison in the workbook.

• Supplementary file 3. Full motif enrichment and accessibility results. Excel workbook containing the full results tables generated by HOMER and chromVAR. The HOMER results contain transcription factor binding motif (TFBM) enrichment statistics for peaks that show an increase in accessibility

from 0 to 3 hr post amputation (hpa) relative to peaks that did not show an increase in accessibility from 0 to 3 hpa. The chromVAR results contain the variability scores for all TFBMs considered in the analysis.

- Supplementary file 4. Primer sequences. Excel worksheet containing the sequences of all primers used in this study.

- Transparent reporting form

### Data availability

All code used in this study is available both as a git repository at https://github.com/cejuliano/jca-zet_regeneration_patterning (copy archived at https://archive.softwareheritage.org/swh:1:rev:6e9b9067f0adf1bd223c5ee0194d75847fa18321/) and on Dryad at https://doi.org/10.25338/B8S612. FASTQ files of raw ATAC-seq and RNA-seq reads, expression matrices for ATAC-seq and RNA-seq reads mapped to the *Hydra* 2.0 genome reference, consensus peak files, and bigwig genome tracks of individual and pooled ATAC-seq replicates are available through the Gene Expression Omnibus under the accession GSE152994. The *Hydra* 2.0 genome gene model IDs associated with the gene names used throughout this study are provided in Figure 1—figure supplement 8. Full differential gene expression results are available in Supplementary files 1 and 2. Motif enrichment and variability results are available in Supplementary file 3.

The following datasets were generated:

| Author(s) | Year | Dataset title | Dataset URL | Database and Identifier |
|---|---|---|---|---|
| Cazet JF, Juliano CJ | 2020 | Oral Regeneration Is the Default Pathway Triggered by Injury in *Hydra* | https://www.ncbi.nlm.nih.gov/geo/query/acc.cgi?acc=GSE152994 | NCBI Gene Expression Omnibus, GSE152994 |
| Cazet JF, Juliano CJ | 2020 | Oral Regeneration Is the Default Pathway Triggered by Injury in Hydra | https://doi.org/10.25338/B8S612 | Dryad Digital Repository, 10.25338/B8S612 |

The following previously published datasets were used:

| Author(s) | Year | Dataset title | Dataset URL | Database and Identifier |
|---|---|---|---|---|
| Siebert S, Cazet JF, Farrell JA | 2018 | Stem cell differentiation trajectories in Hydra resolved at single cell resolution | https://www.ncbi.nlm.nih.gov/geo/query/acc.cgi?acc=GSE121617 | NCBI Gene Expression Omnibus, GSE121617 |
| Wenger Y, Galliot B | 2019 | Generic and context-dependent gene modulations during Hydra whole body regeneration | https://www.ncbi.nlm.nih.gov/geo/query/acc.cgi?acc=GSE111534 | NCBI Gene Expression Omnibus, GSE111534 |

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
