## [Decision Letter]

**Acceptance summary:**

This clearly written manuscript from Cazet et al. sets out to define the molecular logic of positional specification in *Hydra* using RNAseq and ATACseq to profile transcriptome/chromatin accessibility changes that occur within the first 12 hours after injury. Although the initial response appears similar, the oral/aboral injuries begin to diverge by 8 hrs after injury, with several components of the Wnt pathway more highly upregulated in oral vs aboral regeneration. As a potential mechanism for this divergence, TCF binding sites appear more open in oral vs aboral regeneration. A TCF inhibitor previously shown to block regeneration prolongs high expression of transcription factors fos, jun, creb, and AP-1, and there are CRE response elements upstream of these genes. In the final, very clever experiment of the paper, the authors prolong wound signaling in the presence/absence of tissue organizers and find that increased wound signaling increases the prevalence of head regeneration at wound sites. Overall, this paper makes an important contribution to our understanding of regeneration by providing a clear molecular explanation for an older descriptive model and confirms a general theme in metazoan regeneration that initial signaling is general, but that it gets refined by signals from the rest of the tissue.

**Decision letter after peer review:**

[Editors’ note: the authors submitted for reconsideration following the decision after peer review. What follows is the decision letter after the first round of review.]

Thank you for submitting your work entitled "Oral Regeneration Is the Default Pathway Triggered by Injury in *Hydra*" for consideration by *eLife*. Your article has been reviewed by 3 peer reviewers, and the evaluation has been overseen by a Reviewing Editor and a Senior Editor. The reviewers have opted to remain anonymous.

Our decision has been reached after consultation between the reviewers. Based on these discussions and the individual reviews below, we regret to inform you that your work will not be considered further for publication in *eLife*.

The three reviewers agreed that this manuscript reports interesting observations and is a nice gene expression resource with some associated hypotheses but lacks the level of experimental insight that would provide a major advance in the field. The descriptive genomics/transcriptomics data sets are valuable but should be experimentally tested. The reviewers concluded that without further experiments to test specific requirements or regulatory logic, the present work is primarily adding correlative gene expression evidence to a previously known phenotypic result which dampened their enthusiasm. In addition, the conceptual interpretation of the results clings unnecessarily closely to the Gierer/Meinhardt framework and important findings since the publication of this framework have not been taking into consideration.

Reviewer #1:

This very clearly written manuscript from Cazet et al. sets out to define the molecular logic of positional specification in *Hydra* using RNAseq and ATACseq to profile transcriptome/chromatin accessibility changes that occur within the first 12 hours after injury. Although the initial response appears similar, the oral/aboral injuries begin to diverge by 8 hrs after injury, with several components of the Wnt pathway more highly upregulated in oral vs aboral regeneration. As a potential mechanism for this divergence, TCF binding sites appear more open in oral vs aboral regeneration. A TCF inhibitor previously shown to block regeneration prolongs high expression of transcription factors fos, jun, creb, and AP-1, and there are CRE response elements upstream of these genes. In the final, very clever experiment of the paper, the authors prolong wound signaling in the presence/absence of tissue organizers, and find that increased wound signaling increases the prevalence of head regeneration at wound sites. Overall, this paper makes an important contribution to our understanding of regeneration by providing a clear molecular explanation for an older descriptive model, and confirms a general theme in metazoan regeneration that initial signaling is general, but that it gets refined by signals from the rest of the tissue.

1. The mechanisms are nicely described with bioinformatics, but functional experiments would really add significantly to the manuscript. For example, if the impaling experiment in Figure 5 is done in the presence of the TCF inhibitor, does it blunt the aberrant head regeneration? The increase in Wnt expression stimulated regardless of the presence of head/foot organizer tissue is an unsatisfying explanation, because the Wnts obviously increase dramatically in response to the injury. Perhaps this single timepoint is not the appropriate one to test; RT-qPCR timecourses may provide more resolution and identify some differences.

2. In the other *Hydra* strains that don't regenerate heads at the wound site, is there less Wnt/wntless/brachyury expression? Characterizing Wnt expression in these strains in response to injury would be both interesting, potentially useful, and may further strengthen the model.

3. The increased accessibility of TCF binding sites in head regeneration (Figure 2G) is beautiful, but would be more informative and compelling if placed in context. For example, Gehrke et al. included a comprehensive analysis of motifs in Hofstenia.

4. Use of the TCF inhibitor throughout assumes that the effect shown in the Gonsalves paper causes a similar effect in this lab, and is of similar penetrance. It is important to verify that the TCF inhibitor prevents both head and foot regeneration to provide confidence in the inhibition.

Reviewer #2:

Owing to its spectacular ability to regenerate entire polyps from small tissue fragments, *Hydra* is a historically important model for animal regeneration. When polyps are bisected along their oral-aboral axis, an initial wound response occurs and is followed by either oral or aboral regeneration. In "Oral Regeneration Is the Default Pathway Triggered by Injury in *Hydra*," Cazet and Juliano investigate the transcriptional mechanisms underlying this differential oral-aboral response to injury in *Hydra*. The authors utilize RNA-seq and ATAC-seq libraries to look at gene expression changes on a global scale, as well as drug inhibition studies to investigate the role of canonical Wnt signaling in the regenerative response. Their key finding is that Wnt signaling is activated as a component of the generalized injury response but only maintained in regenerating oral fragments, implying that aboral regeneration must entail a context specific repressive effect.

Overall, the genomic aspects of the work (RNA-seq and ATAC-seq) appear technically strong. My chief concerns, while not an expert in the field, surround the conceptual novelty and related to that, the degree of advance provided by the manuscript.

1. Conceptual novelty/degree of advance. The authors should make a stronger effort to differentiate this study from others, particularly Wegner et al. 2019 which really covers much of the same ground, defines the same oral-aboral transcriptional divergence time, and comes to some conceptually similar conclusions regarding the general injury response. It would also help general readers to point more explicitly toward the aspects of this work that are unique or different from what was previously known (eg Petersen et al. MBE 2015; Gufler et al. 2018 and the extensive literature on Wnt signaling in *Hydra* oral patterning/regeneration). As it reads now, the paper is well executed and may reflect a useful resource of -omics data for *Hydra* specialists, but the results are largely incremental and confirmatory of the long-known role of Wnt signaling in injury responses, regeneration, and body patterning throughout animals (eg Whyte et al. 2012).

2. Functional requirements for Wnt signaling. Gufler et al. (2018) utilized iCRT14 treatments to directly demonstrate a requirement for Wnt pathway activity in aboral regeneration. Here the authors use a similar paradigm to characterize the transcriptional changes that result from inhibiting Wnt signaling in oral and aboral fragments. Without further experiments to test specific requirements or regulatory logic, the present work is primarily adding correlative gene expression evidence to a previously known phenotypic result.

3. Wnt activation. To better understand how Wnt is regulated during regeneration, the authors looked for binding sites for known injury responsive TFs is cis regulatory regions of Wnt pathway components, and then correlated these sites with changes in chromatin accessibility during regeneration. A large number of such sites are identified and used to implicate bZIP TFs as "likely playing a role", but no validation is provided (either loss of function studies or even transgenic reporter constructs with and without the elements proposed to drive injury-responsive expression). Without this level of experimentation, the present work is a great resource of gene expression data and possible new hypothesis but lacks the experimental component that would represent a major advance in the field.

Reviewer #3:

This manuscript examines the interplay between wound polarity and gene expression during *Hydra* regeneration. Key points include i) the initial activation of Wnt pathway component expression irrespective of wound polarity, ii) the enrichment of bZIP binding sites in the promoters of multiple Wnt genes; iii) prolonged expression of early wound response transcription factors upon interference with Wnt signaling and iv) the induction of hypostome-like features in the *Hydra* foot upon experimental prevention of wound closure. Overall, the authors interpret their data as confirmation of a Gierer/Meinhardt model that was proposed 40 years ago.

Although the combination of RNAseq and ATACseq provides interesting glimpses of the underlying gene regulatory mechanisms, much of the data and analyses appear somewhat premature. This is especially the case for the latter parts of the manuscript that go beyond what is already known from published RNAseq studies (e.g., common wound response; upregulation of Wnt pathway components early during regeneration, timing of head organizer specification). Moreover, the conceptual interpretation of the results and the model the authors arrive at is overly narrow in parts. Altogether, this manuscript does report interesting observations, but would require extensive revisions to be suitable for publication.

1. The authors need to make their data more accessible.

Figures 1 and 3 require heat maps to illustrate i) the specific expression changes of all the IRTs and IRPs and ii) the identity of the underlying genes. This is essential for making the data broadly accessible to the community. Further, broadening the text perspective beyond Wnt & usual suspects would make the manuscript a lot more engaging, i.e., by naming other genes with interesting expression dynamics or enrichment analyses of Wnt pathway components versus all differentially expressed genes or the genes that become upregulated upon Wnt inhibition in 4.

2. The manuscript takes an overly simplistic view of Wnt signaling.

a. Expression of Wnt signaling components does not equate pathway activation- this assumption is made at multiple points in the text and needs to be corrected.

b. Changes in the accessibility of TCF binding sites does not equate Wnt signaling. The authors disregard that TCF can bind either bind βCatenin or transcriptional repressors (e.g., Groucho). Hence the "opening" of TCF binding sites alone does not signify the activation of Wnt signaling, as it could merely reflect the establishment of Wnt sensitivity or even Wnt-independent gene repression. The authors also cite the molecular mechanism of action of iCRT14 in this context, which is relevant to the interpretation of their results.

c. The authors appear to entirely disregard the many different branches of Wnt signaling that could conceivably result in very different signals in spite of common expression of a subset of transduction components.

Alltogether, these considerations make the "default activation of the head program" a very tentative premise at best. Without a read-out of canonical Wnt signaling activity and comparison of signaling activity between tail and foot regenerating wounds, it is equally possible that the wound type independent expression of Wnt only establishes the potential for future pathway activity, with the actual initiation of signaling requiring a context-dependent trigger.

3. The conceptual interpretation of the results clings unnecessarily closely to the Gierer/Meinhardt framework and remains insufficiently tested.

a. Important findings since Gierer/Meinhardt are not cited or introduced, including for example the suggestion of wound-type specific responses already within the first hour after injury (Chera et al., 2009) or the recent report of dynamic, actin-mediated intrinsic tissue polarization (Livshits et al., 2017). These papers and their key findings need to be cited in the introduction and they need to be discussed in the context of an evaluation of the author's model in the discussion. Moreover, the authors missed that a wound-context independent activation of Wnt pathway transcription also occurs in planarians. This paper needs to be cited (Stuckemann et al., 2017) and the interpretation of wound type independent Wnt signaling in this study might also be of interest in contemplating the *Hydra* data.

b. Necessary tests of the present model include examination of the assumption that the inhibition of Wnt signaling is really sufficient for the conversion of an established head organizer into a foot organizer. This could be easily done by treating regenerating *Hydra* with iCRT14 post 8 h or when the putative Wnt signaling requirement in the initial wound response has passed. Further interesting validations/tests could include the examination of position-dependence of impalement-induced head formation along the body axis (does the efficiency increase with the distance to existing or former "inhibitor sources"? Are any injuries capable of inducing feet?) or qPCR comparisons of Wnt induction efficiency between the two *Hydra* strains.

c. The model disregards the organizing influence of the *Hydra* foot. As is, one might expect that pieces resulting from bisection and additional foot amputation might regenerate double heads (due to absence of the "source" of the long-range inhibitor). This is not observed. Also, the close spacing of the injury-induced double heads in Figure 5 is a problem for the Gierer/Meinhardt framework, as the inhibitor emanating from each head would be expected to interfere with the formation/maintenance of the other head organizer due to their close spacing.

4. Validation of the importance of bZIP binding motifs in Wnt promoter regions.

As is, this section reads very much like a "just so" story. First, the authors have to analyze the probability of bZIP binding motif occurrence per kb in the *hydra* genome. Second, they should examine the occurrence of the motif in non wound-sensitive promoters. Third, they have to demonstrate significant enrichment of the motif in wound-induced genes. The strong emphasis on the importance of the bZip binding motifs in the present manuscript further necessitates experimental validation- either in the manuscript itself or by citing published data in a peer-reviewed journal.

[Editors’ note: further revisions were suggested prior to acceptance, as described below.]

Thank you for submitting your article "Generic Injuries Are Sufficient to Induce Ectopic Wnt Organizers in *Hydra*" for consideration by *eLife*. Your article has been reviewed by 1 peer reviewer, and the evaluation has been overseen by a Reviewing Editor and Edward Morrisey as the Senior Editor. The reviewers have opted to remain anonymous.

The reviewers have discussed the reviews with one another and the Reviewing Editor has drafted this decision to help you prepare a revised submission.

Summary:

This clearly written manuscript from Cazet et al. sets out to define the molecular logic of positional specification in *Hydra* using RNAseq and ATACseq to profile transcriptome/chromatin accessibility changes that occur within the first 12 hours after injury. Although the initial response appears similar, the oral/aboral injuries begin to diverge by 8 hrs after injury, with several components of the Wnt pathway more highly upregulated in oral vs aboral regeneration. As a potential mechanism for this divergence, TCF binding sites appear more open in oral vs aboral regeneration. A TCF inhibitor previously shown to block regeneration prolongs high expression of transcription factors fos, jun, creb, and AP-1, and there are CRE response elements upstream of these genes. In the final, very clever experiment of the paper, the authors prolong wound signaling in the presence/absence of tissue organizers and find that increased wound signaling increases the prevalence of head regeneration at wound sites. Overall, this paper makes an important contribution to our understanding of regeneration by providing a clear molecular explanation for an older descriptive model and confirms a general theme in metazoan regeneration that initial signaling is general, but that it gets refined by signals from the rest of the tissue.

The authors addressed many of the previous critiques. The revised manuscript is substantially rewritten and several additional experiments and figures were added.

Revisions:

1. The authors now added acridine orange stainings to demonstrate that apoptosis is induced in head and foot regenerates, which is in contrast to what Chera et al. published in 2009. However, the acridine orange stainings have quite a bit of background and variability (and they are not quantified). As the Chera et al., data also looks convincing, the authors are encouraged to repeat a TUNEL staining that Chera et al., used. This would allow a better comparison between the differing results.

2. The literature describing the generic injury response that gradually polarizes that has been extensively described in planarians should be better cited. For example, Gurley et al., Dev Biol 2010 should be cited.

3. Please arrange panels in Figure 3 according to how they are cited to facilitate easy reading of the manuscript.

---

## [Author Response]

[Editors’ note: The authors appealed the original decision. What follows is the authors’ response to the first round of review.]

Reviewer #1:This very clearly written manuscript from Cazet et al. sets out to define the molecular logic of positional specification in Hydra using RNAseq and ATACseq to profile transcriptome/chromatin accessibility changes that occur within the first 12 hours after injury. Although the initial response appears similar, the oral/aboral injuries begin to diverge by 8 hrs after injury, with several components of the Wnt pathway more highly upregulated in oral vs aboral regeneration. As a potential mechanism for this divergence, TCF binding sites appear more open in oral vs aboral regeneration. A TCF inhibitor previously shown to block regeneration prolongs high expression of transcription factors fos, jun, creb, and AP-1, and there are CRE response elements upstream of these genes. In the final, very clever experiment of the paper, the authors prolong wound signaling in the presence/absence of tissue organizers, and find that increased wound signaling increases the prevalence of head regeneration at wound sites. Overall, this paper makes an important contribution to our understanding of regeneration by providing a clear molecular explanation for an older descriptive model, and confirms a general theme in metazoan regeneration that initial signaling is general, but that it gets refined by signals from the rest of the tissue.

We thank the reviewer for recognizing the important contribution of our work. One misunderstanding we would like to point out: the CRE elements are found in the regulatory regions of the Wnt genes, not in the regulatory regions of the bZIP TFs (see Figure 4). The manuscript has been substantially re-written, so we hope that this point is now clearer.

1. The mechanisms are nicely described with bioinformatics, but functional experiments would really add significantly to the manuscript. For example, if the impaling experiment in Figure 5 is done in the presence of the TCF inhibitor, does it blunt the aberrant head regeneration?

We agree that the inclusion of functional experiments can greatly enhance the value of a descriptive dataset. As such, we’d like to highlight the functional data that we included in our original submission, including the iCRT14-treated RNA-seq and ATAC-seq datasets presented in figure 3 and the tissue manipulation experiments presented in figure 5. These data allow us to functionally validate our use of chromatin accessibility data to quantify TCF transcriptional activity (figure 3A), to identify novel regulatory functions for TCF during regeneration (Figure 3B-H), and to demonstrate the sufficiency of injury to induce ectopic Wnt organizers (Figure 5).

We also agree that the addition of an experiment to determine if injury-induced ectopic head formation is TCF-dependent would improve the manuscript. Therefore we performed the experiment suggested by the reviewer and present the results in Figure 5—figure supplement 1A. We found that iCRT14 significantly reduced the number of ectopic tentacles formed after prolonged transverse impalement, thus demonstrating that the phenotype is TCF-dependent.

The increase in Wnt expression stimulated regardless of the presence of head/foot organizer tissue is an unsatisfying explanation, because the Wnts obviously increase dramatically in response to the injury. Perhaps this single timepoint is not the appropriate one to test; RT-qPCR timecourses may provide more resolution and identify some differences.

It is a valid and useful point that the qPCR data we present from impaled tissue does not shed light on why the presence or absence of pre-existing organizers influences the outcome of prolonged impalement experiments. However, in our original manuscript, it was not our intention to use these data to address that particular question. Rather, we were aiming to demonstrate that despite the different outcomes, the initial response was nonetheless the same. This was in service of our broader argument that injury-induced expression of Wnt pathway components is not sensitive to signals from the surrounding tissue context during the generic phase of the injury response (even if that generic phase is prolonged to last for 12 hours).

In interpreting the results of the impalement experiments, we worked under the assumption that the observation that one treatment condition resulted in ectopic head organizers and the other did not was sufficient evidence that the two conditions resulted in differences in Wnt signaling activity. Essentially, our position is that the extensive research on the role of Wnt signaling in head specification in *Hydra*  renders the alternative hypothesis (that head organizer formation did not result from differential Wnt signaling activity) implausible. We believe that the fact that aspects of our rationale were not made fully explicit in the original text contributed to the lack of clarity on this point. We hope that the extensive changes we made to our discussion of the impalement experiments help to make understanding our rationale more straightforward.

Although we still maintain that head formation can be taken as a reliable proxy for Wnt signaling activity and Wnt pathway gene expression, we nonetheless recognize that investigating later time points using qPCR could serve as a valuable validation of our underlying assumptions. We therefore characterized Wnt pathway gene expression four days after impalement in animals that did form ectopic heads and those that did not (Figure 5—figure supplement 1H). We found that ectopic heads expressed Wnt pathway genes at significantly higher levels than tissue lacking ectopic heads. This therefore demonstrates that pre-existing organizers downregulated Wnt pathway genes at some point following impalement, although it is unclear when or how.

2. In the other Hydra strains that don't regenerate heads at the wound site, is there less Wnt/wntless/brachyury expression? Characterizing Wnt expression in these strains in response to injury would be both interesting, potentially useful, and may further strengthen the model.

We agree that this is an important question to address. In our revised manuscript, we have included qPCR data showing that 12 hours of impalement resulted in the significant upregulation of Wnt pathway genes in both 105 and AEP strain animals (Figure 5—figure supplement 1F,G). Thus, while ectopic head formation is strain specific (we observe it in Basel and 105 animals but not AEP animals) the injury induced expression of Wnt pathway components following puncture wounds is not.

3. The increased accessibility of TCF binding sites in head regeneration (Figure 2G) is beautiful, but would be more informative and compelling if placed in context. For example, Gehrke et al. included a comprehensive analysis of motifs in Hofstenia.

We think that presenting a systematic analysis of binding site accessibility would indeed provide a useful context for our discussion of the accessibility dynamics of TCF binding sites specifically. We wish to point out that we did present such a systematic analysis in Figure 4A of our original manuscript. However, because we presented these results quite a bit after our discussion of TCF binding site accessibility, it may not have effectively provided the context requested by the reviewer. In our revised manuscript, we now present our systematic analysis of motif accessibility in Figure 1I and explicitly discuss TCF in the context of the broader trends in transcription factor binding motif accessibility during regeneration.

4. Use of the TCF inhibitor throughout assumes that the effect shown in the Gonsalves paper causes a similar effect in this lab, and is of similar penetrance. It is important to verify that the TCF inhibitor prevents both head and foot regeneration to provide confidence in the inhibition.

We recognize the importance of demonstrating the reproducibility previous findings in the literature and we appreciate the suggestion that we include such data in our manuscript (Note: We assume the reviewer intended to refer to the paper by Gufler et al. which first reported the block of *Hydra*  head and foot regeneration, not the paper by Gonsalves et al. which initially developed and biochemically characterized iCRT14 in cultured cells). Before using iCRT14 in our study, we did do these validations, but we did not present them in our original manuscript. We have rectified this oversight and added these experiments to our revised manuscript demonstrating that iCRT14 significantly inhibited both head and foot regeneration in our hands (Figure 3—figure supplement 1A,B), thus recapitulating previously reported findings.

Reviewer #2:Owing to its spectacular ability to regenerate entire polyps from small tissue fragments, Hydra is a historically important model for animal regeneration. When polyps are bisected along their oral-aboral axis, an initial wound response occurs and is followed by either oral or aboral regeneration. In "Oral Regeneration Is the Default Pathway Triggered by Injury in Hydra," Cazet and Juliano investigate the transcriptional mechanisms underlying this differential oral-aboral response to injury in Hydra. The authors utilize RNA-seq and ATAC-seq libraries to look at gene expression changes on a global scale, as well as drug inhibition studies to investigate the role of canonical Wnt signaling in the regenerative response. Their key finding is that Wnt signaling is activated as a component of the generalized injury response but only maintained in regenerating oral fragments, implying that aboral regeneration must entail a context specific repressive effect.Overall, the genomic aspects of the work (RNA-seq and ATAC-seq) appear technically strong. My chief concerns, while not an expert in the field, surround the conceptual novelty and related to that, the degree of advance provided by the manuscript.1. Conceptual novelty/degree of advance. The authors should make a stronger effort to differentiate this study from others, particularly Wegner et al. 2019 which really covers much of the same ground, defines the same oral-aboral transcriptional divergence time, and comes to some conceptually similar conclusions regarding the general injury response. It would also help general readers to point more explicitly toward the aspects of this work that are unique or different from what was previously known (eg Petersen et al. MBE 2015; Gufler et al. 2018 and the extensive literature on Wnt signaling in Hydra oral patterning/regeneration). As it reads now, the paper is well executed and may reflect a useful resource of -omics data for Hydra specialists, but the results are largely incremental and confirmatory of the long-known role of Wnt signaling in injury responses, regeneration, and body patterning throughout animals (eg Whyte et al. 2012).

We recognize that the way our manuscript was written may not have provided readers with sufficient context to understand our work’s contribution to the regeneration field. In our revised manuscript, we have entirely changed the way in which we frame our data. The introduction in the revised text now states more clearly the current gaps in knowledge in the *Hydra* regeneration literature, and the Results section now includes more specific explanations of the novelty of our findings. We hope that these changes will provide the reader with a better understanding of this manuscript’s contribution to regeneration biology.

A critical point that we did not sufficiently stress in our original manuscript is that a generic role for Wnt signaling during an initial context-independent injury response in *Hydra* is not long known or well established. Indeed, the best known molecular model of *Hydra* regeneration claims that Wnt signaling is exclusively activated during head regeneration within 1 hour post amputation (Chera et al., 2009). In addition, recent work by Ramirez et al. (2020) reports that Wnt signaling is rapidly activated during tail, but not head regeneration in aceols. Our work is the first to document the upregulation of Wnt ligands and other Wnt signaling components in non-regenerative injuries and during foot regeneration in *Hydra*. We are also the first to characterize TCF activity during Hydra regeneration. These findings significantly impact not only our understanding of cnidarian regeneration, but also our understanding of the evolution of the metazoan wound response. In our revised manuscript, we now more clearly explain the importance and novelty of our findings on Wnt signaling during regeneration. In particular, we now extensively discuss both the Chera et al. and Ramirez et al. publications in our introduction and discussion. Furthermore, we provide new data that directly refutes the claim made in the Chera et al. paper that injury apoptosis occurs in oral-facing but not aboral-facing amputation injuries (Figure 1J-M, Figure 1—figure supplement 5).

The recently released pre-print from Wenger et al. (2019) does contain an RNA-seq dataset that overlaps with our own. Based on this RNA-seq data, the authors report the existence of an initial context independent transcriptional response to injury and estimate that transcription during head and foot regeneration diverge between 4 and 8hpa. These findings are highly consistent with the conclusions we draw in figure 1(C-H) of our manuscript. However, we wish to emphasize that our study goes far beyond these two observations and makes numerous additional contributions that are wholly distinct from the work of Wenger et al.

1. Wenger et al. do not report the initial context-independent upregulation of Wnt signaling components during the context-independent injury response in their paper. However, we did find that this phenomenon is supported by their data after performing our own analysis.

2. They do not characterize chromatin remodeling and are thus far more limited in their ability to shed light on transcriptional regulation during head and foot regeneration.

3. They do not explore potential mechanisms that drive the transcriptional divergence of head and foot regeneration, nor do they explore potential mechanisms driving injury-responsive Wnt expression.

4. Their study does not include the functional experiments included in our study (figures 3 and 5)

The Gufler et al. (2018) paper was the first to report that TCF is required for both head and foot regeneration in *Hydra* using the TCF/β-catenin inhibitor iCRT14. In addition, they show that TCF inhibition prevents the downregulation of 8 head regeneration-specific genes during foot regeneration. Based on these findings, they proposed that Wnt signaling may play a role during foot regeneration. In our work, we extend these initial findings by comprehensively characterizing chromatin accessibility and transcript abundance during head and foot regeneration in the presence of iCRT14. With these data we find that TCF is required for the divergence in expression of all transcripts during head and foot regeneration, not just the 8 head-specific genes reported in Gufler et al. In addition, we find that TCF is required for the proper downregulation of the injury response and for the upregulation of head and foot specific transcripts. These inhibitor-dependent changes in transcription demonstrate a causal relationship, and allow us to better understand TCF function during regeneration. Importantly, these findings are wholly novel and are not foregone conclusions that can be deduced from the work of Gufler et al. Finally, by using our chromatin accessibility data to characterize TCF transcriptional activation during regeneration, we are able to get a readout of Wnt signaling activity that is comparable to popular TCF-based reporter assays (e.g. TOPFLASH). Such a readout, in conjunction with our transcriptomic data, gives us significant insight into Wnt signaling activity during regeneration that was not possible given the limited molecular data presented in Gufler et al. In our revised manuscript, we now more clearly delineate which of the results from our iCRT14-treated RNA-seq and ATAC-seq data are recapitulations of previous findings (Figure 3 – —figure supplement 1) and which are novel to this study (Figure 3, Figure 3—figure supplement 2).

The Petersen et al. (2015) paper included a comprehensive transcriptomic characterization of head regeneration in *Hydra*. Because this study exclusively focused on head regeneration, it does not shed light on how head and foot regeneration diverge, which is the primary concern of our paper. In addition, all the points outlined above for Wenger et al. also apply.

While there are numerous other studies covering Wnt signaling’s role in *Hydra* patterning, they do not characterize the pathway outside the context of head specification during regeneration and homeostasis. Therefore, our work looking at Wnt signaling during foot regeneration and in response to non-amputation injuries is novel.

2. Functional requirements for Wnt signaling. Gufler et al. (2018) utilized iCRT14 treatments to directly demonstrate a requirement for Wnt pathway activity in aboral regeneration. Here the authors use a similar paradigm to characterize the transcriptional changes that result from inhibiting Wnt signaling in oral and aboral fragments. Without further experiments to test specific requirements or regulatory logic, the present work is primarily adding correlative gene expression evidence to a previously known phenotypic result.

Please refer to our response to the first comment for the ways in which our study is distinct from the work presented by Gufler et al. Briefly, these are the novel functional results arising from our iCRT14 experiments:

1. TCF is required for the downregulation of only a small subset of head specific factors during foot regeneration, which includes the ligands *wnt3* and *wnt9/10c*.

2. TCF is required to downregulate aspects of the generic wound response

3. TCF is required to upregulate foot-specific transcripts during foot regeneration

4. TCF is required for the upregulation of a subset of head specific transcripts

5. TCF is required for all context-specific expression during head and foot regeneration

In addition, we would like to note that the functional data presented in figure 5 demonstrate that non-amputation injuries are sufficient to activate Wnt signaling and induce the formation of secondary axes. These findings are entirely unique to our work.

3. Wnt activation. To better understand how Wnt is regulated during regeneration, the authors looked for binding sites for known injury responsive TFs is cis regulatory regions of Wnt pathway components, and then correlated these sites with changes in chromatin accessibility during regeneration. A large number of such sites are identified and used to implicate bZIP TFs as "likely playing a role", but no validation is provided (either loss of function studies or even transgenic reporter constructs with and without the elements proposed to drive injury-responsive expression). Without this level of experimentation, the present work is a great resource of gene expression data and possible new hypothesis but lacks the experimental component that would represent a major advance in the field.

We wholeheartedly agree that loss of function and reporter experiments would be ideal approaches for directly testing our hypothesis that Wnt signaling components are directly regulated by injury-responsive bZIP transcription factors. We have either attempted, or are currently attempting both types of experiments; however, we face significant technical hurdles presented by our model system. The *Hydra* research community is small, and lacks many of the tools that are readily available in more widely used model systems. Our lab is working on improving those tools, but we have only limited resources to address these significant challenges.

Currently, the best established protocol for gene knockdown in *Hydra* is to create a transgenic line that constitutively expresses a RNAi hairpin directed against the gene of interest. Although we have attempted to establish such knockdown lines for multiple injury-responsive bZIP transcription factors, we have failed to generate any KD positive transgenic lines. This can occur when a gene of interest is essential. Therefore perturbing the function of these transcription factors will likely first require the development of an inducible knockdown system in *Hydra*.

The only available method for transgenesis in *Hydra* (random insertion of injected circular DNA) makes performing reporter experiments difficult. We are currently unable to control where the injected plasmid breaks or where it is inserted in the genome. Reporter experiments therefore require a large number of injections for each construct being tested. We are currently attempting to generate lines to test the function of putative bZIP binding sites in the Wnt9/10c promoter, but this work is slow and has been hindered by the pandemic. Our plan is to present such data in a follow-up study.

Importantly, while the above experiments are necessary to test the hypothesis that bZIPs directly regulate Wnt components, they are not necessary for the broader conclusion of the paper, which is that injuries induce Wnt signaling outside the context of head amputation. Functional data supporting this conclusion is presented in figure 5.

Reviewer #3:This manuscript examines the interplay between wound polarity and gene expression during Hydra regeneration. Key points include i) the initial activation of Wnt pathway component expression irrespective of wound polarity, ii) the enrichment of bZIP binding sites in the promoters of multiple Wnt genes; iii) prolonged expression of early wound response transcription factors upon interference with Wnt signaling and iv) the induction of hypostome-like features in the Hydra foot upon experimental prevention of wound closure. Overall, the authors interpret their data as confirmation of a Gierer/Meinhardt model that was proposed 40 years ago.Although the combination of RNAseq and ATACseq provides interesting glimpses of the underlying gene regulatory mechanisms, much of the data and analyses appear somewhat premature. This is especially the case for the latter parts of the manuscript that go beyond what is already known from published RNAseq studies (e.g., common wound response; upregulation of Wnt pathway components early during regeneration, timing of head organizer specification). Moreover, the conceptual interpretation of the results and the model the authors arrive at is overly narrow in parts. Altogether, this manuscript does report interesting observations, but would require extensive revisions to be suitable for publication.

We first wish to clarify a few apparent misunderstandings. Regarding point iv, we did not show the induction of hypostome-like features in the foot following prolonged injuries. Rather, we demonstrated that prolonged injuries could induce ectopic heads in body column tissue. We would also like to point out that, with the exception of *β*-*catenin*, our paper is the first to describe the upregulation of Wnt pathway genes during aboral regeneration and in response to non-amputation injuries. While the RNA-seq dataset presented in Wenger et al. does indeed show the upregulation of Wnt pathway genes during foot regeneration, we were only able to make that conclusion by performing our own analysis on their data, as they do not report this finding anywhere in their manuscript. We would also point out that the Wenger et al. data set is from a pre-print manuscript and has not undergone peer review.

We also wish to clarify that we are not claiming to confirm the original Gierer-Meinhardt model as it was proposed in the initial 1972 publication. Rather, our data provides a molecular basis for the observation made after the Gierer-Meinhardt model was proposed that injuring *Hydra* tissue promotes the formation of a new head organizer, which we refer to as the “oral induction model.” We do invoke a Gierer-Meinhardt type of reaction-diffusion system when presenting our model of patterning during regeneration, but our model incorporates numerous observations made after the original Gierer-Meinhardt publication.

Upon reflection, we believe that our decision to focus on the “oral induction model” as a way to frame our paper led to several issues in our manuscript that were highlighted by the reviewer’s comments. In particular, we did not adequately address the ambiguities and gaps in knowledge currently present in the *Hydra* regeneration literature on Wnt activity and Wnt pathway gene expression during head and foot regeneration. We also did not properly convey that resolving the question of Wnt pathway regulation during *Hydra* regeneration has important implications regarding the evolution of regeneration in metazoans, especially given the findings on Wnt regulation in acoels and planarians. We have made a significant effort to rework the text to address these issues. We now extensively discuss both the planarian and acoel regeneration literature in our introduction and discussion. We also directly address the model presented by Chera et al. (2000), both through modifications to the text and through the inclusion of additional experiments (Figure 1J-M, Figure 1—figure supplement 5). Our Results section has also been reworked to more clearly explain the novelty of our findings in the context of the pre-existing literature. In addition, please refer to our response to comment 1 from reviewer 2 for our in-depth explanation of the ways in which our study can be differentiated from other related works.

1. The authors need to make their data more accessible.Figures 1 and 3 require heat maps to illustrate i) the specific expression changes of all the IRTs and IRPs and ii) the identity of the underlying genes. This is essential for making the data broadly accessible to the community.

We agree that transparency is crucial when presenting scientific findings. We would like to highlight that we did make a meaningful effort to make our data accessible to readers. In particular, we included full results tables for the differential analyses we performed in Supplementary files 1 and 2. In addition, we made all code used in this study publicly available upon submission both on GitHub and Dryad. We also deposited both raw and processed data, including fastq files, peak files, read count matrices, and bigwig files on GSE and made these data publicly available upon submission.

We did inadvertently omit the results tables from the single cell structural enrichment analysis that were used to generate the plots in Figure 2A-D. We have now added these tables to Supplementary files 1 and 2. We have also added the tables that were directly used to generate all scatter plots and heatmaps throughout the manuscript as ‘source data’ files in our revised submission. We have also added Supplementary file 3, which includes results from our HOMER motif enrichment analysis and our chromVAR chromatin accessibility analysis.

We do not believe that the inclusion of heatmaps would be an effective way to make our data more accessible, as the number of peaks/transcripts being considered (hundreds to thousands in many cases) would either require that the heatmaps be excessively large or would render the gene labels too small to be legible.

Further, broadening the text perspective beyond Wnt & usual suspects would make the manuscript a lot more engaging, i.e., by naming other genes with interesting expression dynamics or enrichment analyses of Wnt pathway components versus all differentially expressed genes or the genes that become upregulated upon Wnt inhibition in 4.

In writing this paper, we chose to primarily focus our attention on genes that had either been previously characterized in *Hydra* or were directly relevant to the paper’s main focus—the regulation of Wnt signaling during regeneration. Our intention was that readers with a broader curiosity in our data could easily identify other genes of interest using the supplementary tables we provided. This choice was based on the fact that we did not intend to present our work as a resource paper, but rather we sought to focus on a specific set of questions.

However, we would note that figures 1 and 3 are not exclusively focused on Wnt signaling. In figure 1 of our original manuscript, we highlighted the foot-specific transcription factors (*distall-less* and *nk-2*). In figure 3 we highlighted multiple genes that aren’t Wnt pathway components including *prdl-a*, *gremlin-like*, *budhead*, *creb*, *fos*, and *jun*. Additionally, we highlighted TCF-dependent changes in the accessibility of the injury responsive TRE and Erg transcription factor binding motifs. In our revised manuscript we additionally highlight two novel foot specific TFs that are among the first to be upregulated in a foot regeneration-specific fashion (*gata-3-like* and *foxd2-like*; lines 219-221). We also note that these foot TFs exhibit TCF-dependent expression during regeneration (Figure 3E,F).

2. The manuscript takes an overly simplistic view of Wnt signaling.a. Expression of Wnt signaling components does not equate pathway activation- this assumption is made at multiple points in the text and needs to be corrected.b. Changes in the accessibility of TCF binding sites does not equate Wnt signaling. The authors disregard that TCF can bind either bind βCatenin or transcriptional repressors (e.g., Groucho). Hence the "opening" of TCF binding sites alone does not signify the activation of Wnt signaling, as it could merely reflect the establishment of Wnt sensitivity or even Wnt-independent gene repression. The authors also cite the molecular mechanism of action of iCRT14 in this context, which is relevant to the interpretation of their results.c. The authors appear to entirely disregard the many different branches of Wnt signaling that could conceivably result in very different signals in spite of common expression of a subset of transduction components.Alltogether, these considerations make the "default activation of the head program" a very tentative premise at best. Without a read-out of canonical Wnt signaling activity and comparison of signaling activity between tail and foot regenerating wounds, it is equally possible that the wound type independent expression of Wnt only establishes the potential for future pathway activity, with the actual initiation of signaling requiring a context-dependent trigger.

We agree that it is important to acknowledge that expression of wnt signaling components does not equate to an increase in Wnt signaling activity, which is why we did so in our original manuscript: “Although canonical Wnt signaling components were upregulated during both head and foot regeneration, it was not clear if this resulted in the downstream activation of Wnt responsive transcription.”

We would also like to note that the language at line 163 of our original manuscript did not claim that the upregulation of Wnt pathway transcripts was evidence that the pathway was active, merely that it suggested that such a hypothesis was plausible.

However, we disagree with the reviewer that changes in TCF binding site accessibility can’t be used to characterize Wnt signaling activity. β-catenin/TCF transcriptional activation is driven in part through the recruitment of histone acetyl-transferases (ncbi.nlm.nih.gov/pmc/articles/PMC302022/), which increase chromatin accessibility. Further, transcriptional repressors such as groucho inhibit transcription by decreasing chromatin accessibility through the recruitment of histone de-acetylases

(ncbi.nlm.nih.gov/pmc/articles/PMC316998/). Therefore, an increase in the accessibility of TCF binding sites would be an expected consequence of an increase in TCF transcriptional activation, not repression. Our updated manuscript clarifies this rationale. Furthermore, our iCRT14 data demonstrates that chromatin accessibility changes near TCF binding motifs during regeneration are TCF dependent (Figure 3A), providing validation for this approach.

While we recognize that Wnt signaling is highly complex, we do not believe that this fact prevents us from using our data to characterize canonical Wnt signaling. TOPFLASH—a TCF-based reporter construct—is a well-established and widely used method for assaying Wnt signaling activity. Our ATAC-seq analysis provides information that is comparable to TOPFLASH, as it measures TCF activity via TCF-dependent changes in chromatin accessibility. Therefore, despite the complexities of Wnt signaling, we argue that TCF transcriptional activation can provide a meaningful readout of Wnt signaling activity.

Although we maintain that the above rationale justifies our conclusions regarding Wnt signaling during regeneration, we nonetheless recognize that stating that Wnt signaling is activated during normal foot regeneration is a strong claim that would ideally be validated through multiple sources of evidence (such as characterizing β-catenin protein abundance and localization). We have therefore softened the language we use in discussing our ATAC-seq and RNA-seq data throughout our updated manuscript to be less definitive.

However, our primary conclusion that head formation is the default outcome promoted by injury does not hinge on how we interpret our sequencing data. Rather, the primary evidence for this conclusion is presented in Figure 5, where we demonstrate that prolonged injuries in the body column exclusively leads to ectopic head formation—despite the fact that mid-body column tissue is equally capable of regenerating either a head or a foot. In addition, because Wnt signaling is necessary and sufficient for head formation, our finding that prolonged puncture wounds and aboral amputations induce ectopic heads demonstrates that the generic injury response can induce Wnt signaling. We provide further support for this conclusion with the addition of data showing that iCRT14 significantly reduces the number of ectopic tentacles formed after prolonged injury, demonstrating that the phenotype is TCF-dependent.

3. The conceptual interpretation of the results clings unnecessarily closely to the Gierer/Meinhardt framework and remains insufficiently tested.a. Important findings since Gierer/Meinhardt are not cited or introduced, including for example the suggestion of wound-type specific responses already within the first hour after injury (Chera et al., 2009) or the recent report of dynamic, actin-mediated intrinsic tissue polarization (Livshits et al., 2017). These papers and their key findings need to be cited in the introduction and they need to be discussed in the context of an evaluation of the author's model in the discussion. Moreover, the authors missed that a wound-context independent activation of Wnt pathway transcription also occurs in planarians. This paper needs to be cited (Stuckemann et al., 2017) and the interpretation of wound type independent Wnt signaling in this study might also be of interest in contemplating the Hydra data.

As discussed above, we agree we should have included discussion of Chera et al., 2009 in our manuscript. We have addressed this issue in the revised text and now discuss the findings from Chera et al. throughout the manuscript. We have also added additional experimental evidence that directly addresses aspects of the Chera model (Figure 1J-M, Figure 1—figure supplement 5).

Regarding the data from Livshits et al. (2017), while their findings are interesting, we do not believe their work influences the interpretation of our data. Livshits et al. exclusively focused on how axis orientation is re-established in injury contexts when pre-existing positional information has presumably been lost (i.e. in very small tissue fragments). In contexts where a defined Wnt gradient is present, it is well established that Wnt signaling overrides cytoskeletal orientation. Furthermore, Livshits et al. do not perform any experiments that shed light on how the direction of polarity along the rebuilt axis is established. Their work therefore does not provide insight into the question of head vs. foot decisions in their chosen injury context. Our injury context does not cause the loss of the pre-existing axis and is primarily concerned with head vs. foot specification. Additionally, more recent research has called into question important aspects of the findings presented in the Livshits et al. study (pubmed.ncbi.nlm.nih.gov/32871156). We therefore argue that omission of Livshits et al. from our study does not meaningfully detract from the interpretation of our data.

We agree that findings from the planarian literature have interesting and important implications for the interpretation of our data. While we did cite the planarian literature (see list below), we did omit two important studies (Stuckmann et al., 2017 and Petersen and Reddien, 2009). We have rectified these omissions in our updated manuscript. More broadly, we have modified the text to more extensively discuss findings from bilaterian whole-body regeneration research in both the introduction and discussion.

Planarian papers cited in the original manuscript:

Owlarn and Bartscherer, 2016

Wurtzel et al., 2015

Wenemoser et al., 2012

Owlarn et al., 2017

b. Necessary tests of the present model include examination of the assumption that the inhibition of Wnt signaling is really sufficient for the conversion of an established head organizer into a foot organizer. This could be easily done by treating regenerating Hydra with iCRT14 post 8 h or when the putative Wnt signaling requirement in the initial wound response has passed. Further interesting validations/tests could include the examination of position-dependence of impalement-induced head formation along the body axis (does the efficiency increase with the distance to existing or former "inhibitor sources"? Are any injuries capable of inducing feet?) or qPCR comparisons of Wnt induction efficiency between the two Hydra strains.

We wish to clarify an apparent misunderstanding mentioned by the reviewer. We are not claiming that inhibition of Wnt signaling is sufficient for conversion of an established head organizer into a foot organizer. We do claim that Wnt signaling is transiently activated during early foot regeneration, but in our discussion of this hypothesis we do not claim that this transient activation temporarily results in head organizer formation. Head and foot organizers are self-maintaining and insensitive to inhibitory signals from the surrounding tissue. If a true head organizer formed during foot regeneration it would always trigger ectopic head formation.

We believe that treating regenerating *Hydra* after 8 hours post amputation would likely not be informative, because our molecular data indicates that Wnt signaling has already been repressed in foot regenerates by this time point. In addition, we believe that exploring the effect of position along the oral-aboral axis on ectopic head formation would be largely redundant with results already published in the literature (in particular the MacWilliams studies from 1983), which extensively tested the influence of position on secondary axis formation in *Hydra* using grafting experiments.

To provide additional evidence on the induction of canonical Wnt signaling following aboral-facing amputations, we have added new experimental data in Figure 5E-G of the updated manuscript. In these experiments, we demonstrate that prolonging aboral-facing amputation injuries can induce ectopic head formation at the aboral pole, and that this can be inhibited by tissue undergoing head regeneration at the oral pole. Because Wnt signaling is necessary and sufficient for head organizer formation in *Hydra*, these data demonstrate that aboral injuries can induce Wnt signaling.

c. The model disregards the organizing influence of the Hydra foot. As is, one might expect that pieces resulting from bisection and additional foot amputation might regenerate double heads (due to absence of the "source" of the long-range inhibitor). This is not observed. Also, the close spacing of the injury-induced double heads in Figure 5 is a problem for the Gierer/Meinhardt framework, as the inhibitor emanating from each head would be expected to interfere with the formation/maintenance of the other head organizer due to their close spacing.

The question of why ectopic heads don’t form when both organizers are amputated is indeed interesting and highly relevant to our model. Our inclusion of the experiments in Figure 5E-G addresses this question by looking at this precise injury context. Within the context of our model, the lack of ectopic head formation can be accounted for through two observations: 1) regenerating head tissue gradually increases the amount of inhibition it produces throughout the course of regeneration, 2) head regeneration is faster the closer the amputation site was to the original head organizer. Thus, in cases where both organizers are amputated, the oral-facing amputation injury will recover its inhibitory capacity first, and will subsequently block head formation at the aboral-facing amputation (these points are discussed in the results). The data we present in Figure 5E-G are consistent with this model, as they demonstrate that aboral amputations can induce ectopic heads, but that this can be blocked by tissue undergoing head regeneration at the oral pole.

The close spacing of the injury induced ectopic heads in some transversely animals can also be readily incorporated into our model based on the observation that regenerating heads do not become an effective source of inhibition for some time after they have become committed to head formation (see the discussion in MacWilliams et al., 1983, doi.org/10.1016/0012-1606(83)90325-1). In transverse impalement experiments, both injury sites are at the same position along the oral-aboral axis, and so head organizer formation would be expected to have highly similar kinetics at the two injuries. Essentially, the difference in how long it takes the two organizers to form does not provide enough time for either nascent head to produce sufficient inhibition to block the other. As previously noted, once established, head organizers are self-sufficient and insensitive to inhibitory signals from the surrounding tissue.

4. Validation of the importance of bZIP binding motifs in Wnt promoter regions.As is, this section reads very much like a "just so" story. First, the authors have to analyze the probability of bZIP binding motif occurrence per kb in the hydra genome. Second, they should examine the occurrence of the motif in non wound-sensitive promoters. Third, they have to demonstrate significant enrichment of the motif in wound-induced genes. The strong emphasis on the importance of the bZip binding motifs in the present manuscript further necessitates experimental validation- either in the manuscript itself or by citing published data in a peer-reviewed journal.

We agree that motif enrichment analyses would be an informative addition to the paper. We have in fact performed such analyses and found that CRE sequences are significantly enriched in wound responsive elements. We omitted that work in favor of using the chromVAR analytical approach, but we have now integrated it into our updated manuscript (Figure 4A; Supplementary file 3).

We also agree that, as originally presented, the consideration of only a single injury-responsive motif appears rather arbitrary. In our revised manuscript, we present a systematic analysis of all motifs that exhibited an injury induced increase in chromatin accessibility from 0 to 3 hpa (Figure 4A). Using this systematic approach, we find that while there are other motifs that could plausibly play a role in injury-induced Wnt upregulation, CRE-binding bZIPs remain the most likely candidate regulators.

[Editors’ note: what follows is the authors’ response to the second round of review.]

Revisions:1. The authors now added acridine orange stainings to demonstrate that apoptosis is induced in head and foot regenerates, which is in contrast to what Chera et al. published in 2009. However, the acridine orange stainings have quite a bit of background and variability (and they are not quantified). As the Chera et al., data also looks convincing, the authors are encouraged to repeat a TUNEL staining that Chera et al., used. This would allow a better comparison between the differing results.

We agree that repeating the published TUNEL staining experiment would be a helpful clarification regarding the discrepancy between our manuscript and Chera et al., 2009. Before using acridine orange, we did try TUNEL staining and found that the results were inconsistent and gave high background at early regeneration time points. For this reason, we turned to acridine orange as an alternative approach because it has been previously validated as specifically staining apoptotic cells based on nuclear morphology and was shown to mirror TUNEL staining in *Hydra* (Cikala et al., 1999; Kuznetsov et al., 2002). With acridine orange, we obtain very clear results so this is what we included in the manuscript. In preparing this revision, we did try the TUNEL staining again, but we still could not obtain reasonable results. It should be noted that we saw no evidence of asymmetric TUNEL staining in the experiments that we performed. Rather, the results showed high background and were inconsistent from animal to animal (see Author response image 1). It is important to point out that our RNA-seq results mirror our acridine orange results in that we see symmetric upregulation of genes involved in apoptosis in both head and foot regeneration. Therefore, we have two different lines of evidence that refute the published TUNEL results in Chera et al. In addition, we have added a citation to our manuscript for a recent pre-print article that included TUNEL staining at 2 hpa that also found no evidence of asymmetric apoptosis.

**Author response image 1. sa2fig1:** TUNEL staining gives inconsistent results in uninjured and regenerating tissue. Whole uninjured animals, or animals that were bisected and allowed to regenerate for 1 or 3 hours, were fixed and then stained using the ApopTag TUNEL kit (Σ-Aldrich, S7110). All timepoints and regeneration types, including uninjured controls, exhibited highly variable TUNEL signal..

2. The literature describing the generic injury response that gradually polarizes that has been extensively described in planarians should be better cited. For example, Gurley et al., Dev Biol 2010 should be cited.

We have added citations for Gurley et al. in the introduction, results and discussion. We thank the reviewer for pointing out this omission.

3. Please arrange panels in Figure 3 according to how they are cited to facilitate easy reading of the manuscript.

The panels in Figure 3 are now arranged in the order that they are cited in the text.